# The asymmetrically segregating lncRNA cherub is required for transforming stem cells into malignant cells

Lisa Landskron[1], Victoria Steinmann[1], Francois Bonnay[1], Thomas R Burkard[1], Jonas Steinmann[1], Ilka Reichardt[1], Heike Harzer[1], Anne-Sophie Laurenson[2], Heinrich Reichert[2], Jürgen A Knoblich[1]*

[1]IMBA, Institute of Molecular Biotechnology of the Austrian Academy of Sciences, Vienna, Austria; [2]Biozentrum, University of Basel, Basel, Switzerland

**Abstract** Tumor cells display features that are not found in healthy cells. How they become immortal and how their specific features can be exploited to combat tumorigenesis are key questions in tumor biology. Here we describe the long non-coding RNA cherub that is critically required for the development of brain tumors in *Drosophila* but is dispensable for normal development. In mitotic *Drosophila* neural stem cells, cherub localizes to the cell periphery and segregates into the differentiating daughter cell. During tumorigenesis, de-differentiation of cherub-high cells leads to the formation of tumorigenic stem cells that accumulate abnormally high cherub levels. We show that cherub establishes a molecular link between the RNA-binding proteins Staufen and Syncrip. As Syncrip is part of the molecular machinery specifying temporal identity in neural stem cells, we propose that tumor cells proliferate indefinitely, because cherub accumulation no longer allows them to complete their temporal neurogenesis program.

DOI: https://doi.org/10.7554/eLife.31347.001

*For correspondence:
juergen.knoblich@imba.oeaw.ac.at

Competing interests: The authors declare that no competing interests exist.

## Introduction

Throughout the animal kingdom, stem cells supply tissues with specialized cells. They can do this because they have the unique ability to both replicate themselves (an ability termed self-renewal (*Smith, 2006*)) and to simultaneously generate other daughter cells with a more restricted developmental potential. Besides their role in tissue homeostasis, stem cells have also been linked to tumor formation (*Reya et al., 2001*). They can turn into so-called tumor stem cells that sustain tumor growth indefinitely. The mechanisms that endow tumor stem cells with indefinite proliferation potential are not fully understood.

The fruit fly *Drosophila* has emerged as a genetically tractable system to model tumors in a developmental context and adult tissues (*Gateff, 1978*; *Gonzalez, 2013*) as well as to study naturally occurring tumors (*Salomon and Jackson, 2008*; *Siudeja et al., 2015*). In the developing CNS, neural stem cells, called neuroblasts (NBs) give rise to most neurons and glial cells of the adult fly brain (*Truman and Bate, 1988*). For this, they repeatedly divide into one self-renewing and one differentiating daughter cell (*Kang and Reichert, 2015*; *Neumüller et al., 2011*). Disrupting these asymmetric cell divisions can generate lethal, transplantable brain tumors (*Bello et al., 2006*; *Betschinger et al., 2006*; *Cabernard et al., 2010*; *Janssens and Lee, 2014*; *Knoblich, 2010*; *Lee et al., 2006*; *2006c*; *2006d*). Importantly, the failure to divide asymmetrically has also been linked to tumorigenesis in mammals, particularly in breast cancer (*Cicalese et al., 2009*), myeloid leukemia (*Ito et al., 2010*; *Wu et al., 2007*; *Zimdahl et al., 2014*) and gliomas (*Chen et al., 2014*).

Most *Drosophila* brain tumors originate from the so-called type II neuroblasts (NBIIs) (*Figure 1A*). NBIIs divide asymmetrically into a larger cell that retains NB characteristics and a smaller

**eLife digest** Many biological signals control how cells grow and divide. However, cancer cells do not obey these growth-restricting signals, and as a result large tumors may develop.

Recent experiments have suggested that stem cells – the precursors to the different types of specialized cells found in the body – are particularly important for generating tumors. A stem cell normally divides unequally to form a self-renewing cell and a more specialized cell (often a progenitor cell that will give rise to increasingly specialized cell types). The timing of when the specialization occurs can be key to guiding the ultimately produced cell progenies to their final identity. However, in a tumor cells can retain the ability to self-renew. Ultimately, the resulting 'tumor stem cells' become immortal and proliferate indefinitely. It is not fully understood why this uncontrolled proliferation occurs.

Just like mammals (including humans), fruit flies can develop tumors. Some of the DNA mutations responsible for tumor development were already identified in flies as early as in the 1970s. This has made fruit flies a well-studied model system for uncovering the principle defects that cause tumors to form.

Landskron et al. have now studied the neural stem cells found in brain tumors in fruit flies. Additional DNA mutations were not responsible for these cells becoming immortal. Instead, certain RNA molecules – products that are 'transcribed' from the DNA – were present in different amounts in tumor cells. The RNA that showed the greatest increase in tumor cells is a so-called long non-coding RNA named cherub. This RNA molecule has no important role in normal fruit flies, but is critical for tumor formation.

Landskron et al. found that during cell division cherub segregates from the neural stem cells to the newly formed progenitor cells, where it breaks down over time. Progenitor cells that contain high levels of cherub give rise to tumor-generating neural stem cells. At the molecular level, cherubhelps two proteins to interact with each other: one called Syncrip that makes the neural stem cells take on a older identity, and another one (Staufen) that tethers it to the cell membrane. By restricting Syncrip to a particular location in the cell, cherub alters the timing of stem cell specialization, which contributes to tumor formation.

Overall, the results presented by Landskron et al. reveal a new role for long non-coding RNAs: controlling the localization of the proteins that determine the fate of the cell. They also highlight a critical link between the timing of stem cell development and the proliferation of the cells. Further work is now needed to test whether the same control mechanism works in species other than fruit flies.

DOI: https://doi.org/10.7554/eLife.31347.002

intermediate neural progenitor (INP). Newly formed immature INPs (iINPs) go through a defined set of maturation steps to become transit-amplifying mature INPs (mINPs). After this, a mINP undergoes 3–6 divisions generating one mINP and one ganglion mother cell (GMC) that in turn divides into two terminally differentiating neurons or glial cells (*Bello et al., 2008*; *Boone and Doe, 2008*; *Bowman et al., 2008*).

Similar to mammalian brain progenitors (*Kohwi and Doe, 2013*), *Drosophila* NBs exit proliferation once they complete a specified temporal program during which they generate different types of morphologically distinct progeny (*Homem et al., 2014*; *Liu et al., 2015*; *Maurange et al., 2008*; *Ren et al., 2017*; *Syed et al., 2017*). It is thought that their correct temporal identity requires the RNA-binding proteins IGF-II mRNA-binding protein (Imp) and Syncrip (Syp). During early larval stages, Imp levels are high and Syp levels are low. Over time, Imp expression gradually decreases while the amount of Syp increases. This leads to highly Syp-positive NBs with no detectable Imp at the end of larval development. Manipulating these opposing gradients changes the number and type of neurons made (*Liu et al., 2015*; *Ren et al., 2017*; *Syed et al., 2017*).

During each NBII division, a set of cell fate determinants is segregated into the INP (*Bello et al., 2008*; *Boone and Doe, 2008*; *Bowman et al., 2008*) (*Figure 1A*). Among those are the Notch inhibitor Numb and the TRIM-NHL protein Brain tumor (Brat) (*Bello et al., 2006*; *Betschinger et al., 2006*; *Knoblich et al., 1995*; *Lee et al., 2006d*; *Spana et al., 1995*). Loss of these cell fate

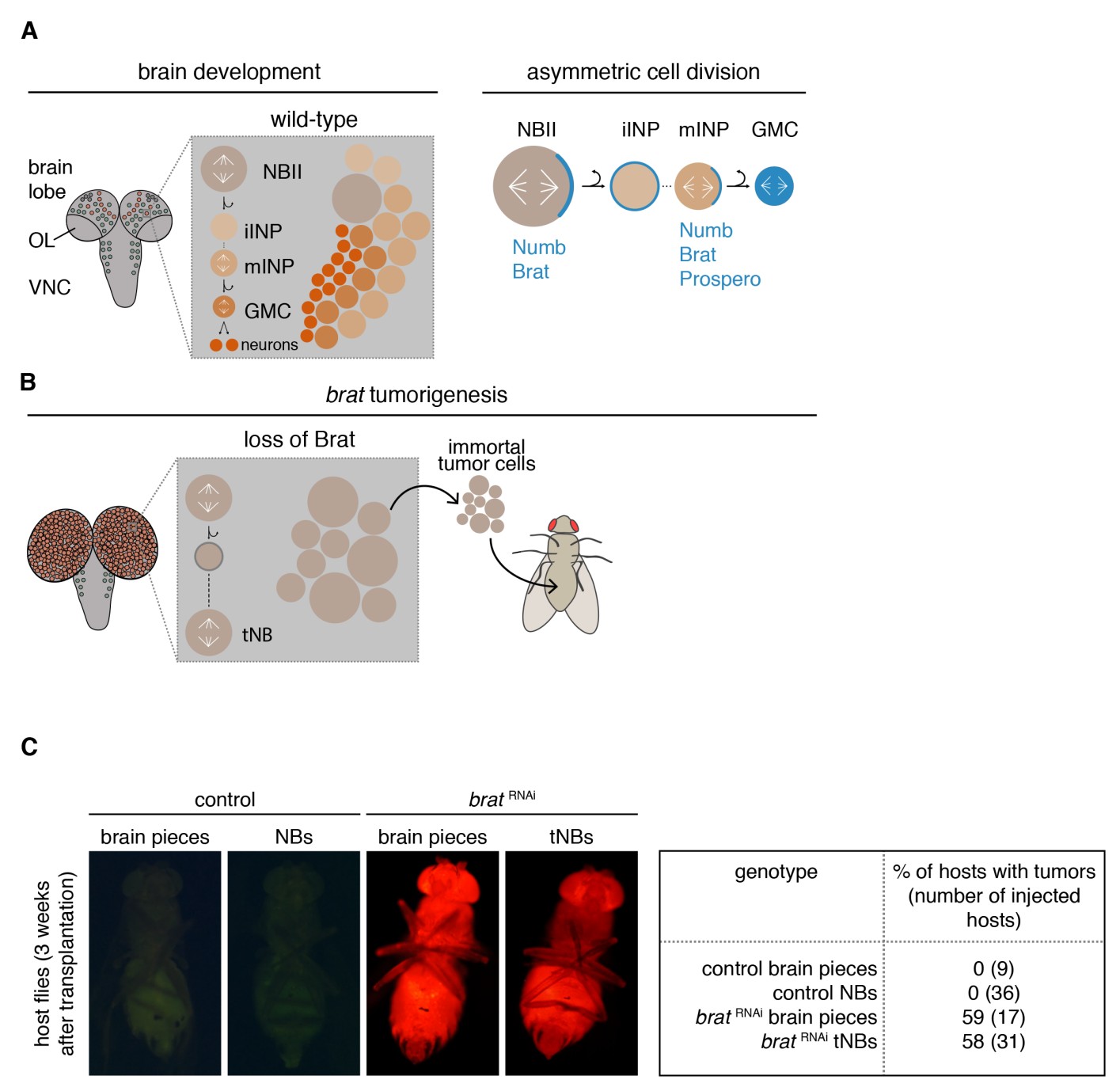

**Figure 1.** *brat* tumor neuroblast possess increased proliferation potential. (**A**) Cartoon depicting a *Drosophila* larval brain (OL optic lobe, VNC ventral nerve cord) harboring different neuroblast populations: mushroom body NBs (grey), type I (NBI, green) and II (NBII, orange) neuroblasts. Close-up shows a NBII lineage (iINP - immature intermediate neural progenitor, mINP - mature INP, GMC - ganglion mother cell) and the typical arrangement of cell types in a NBII clone (left). Proteins (blue) are asymmetrically segregated in NBII and mINP to ensure lineage directionality (right). (**B**) In *brat* mutants, the smaller daughter cell fails to differentiate and after a transient cell cycle block regrows into an ectopic neuroblast (tNB - tumor neuroblast). *brat* tumors continue to grow upon transplantation. (**C**) Representative images of adult host flies injected with FACS-sorted control NBs (GFP+) and *brat* [RNAi] tNBs (RFP+) from third instar larvae. Transplantations of brain pieces served as controls.
DOI: https://doi.org/10.7554/eLife.31347.003

determinants (*Arama et al., 2000*; *Bello et al., 2008*; *Betschinger et al., 2006*; *Gateff, 1978*; *Lee et al., 2006d*; *Wang et al., 2006*) leads to the generation of ectopic NB-like cells at the expense of differentiated brain cells. Formation of malignant brain tumors has also been observed upon the depletion of downstream factors that normally maintain the INP fate (*Eroglu et al., 2014*; *Janssens and Lee, 2014*; *Koe et al., 2014*; *Weng et al., 2010*).

These features make *Drosophila* a model for the stepwise acquisition of tumor stem cell properties. When *numb* or *brat* are inactivated (*Figure 1B*), the smaller NBII progeny fails to establish an INP fate (*Janssens et al., 2014*; *Lee et al., 2006d*) and initially enters a long transient cell cycle arrest (*Bowman et al., 2008*; *Lee et al., 2006d*). Only after this lag period, the smaller cell regrows to a NB-sized cell that has acquired tumor stem cell properties and that we therefore refer to as tumor neuroblast (tNB) (*Bello et al., 2008*; *Betschinger et al., 2006*; *Bowman et al., 2008*; *Lee et al., 2006d*). NBIIs and ectopic tNBs are indistinguishable in terms of markers. Both cell populations are characterized by the expression of self-renewal genes and lack differentiation markers (*Bello et al., 2006*; *Betschinger et al., 2006*; *Lee et al., 2006d*), but nevertheless behave differently. Shortly after entering pupal stages, NBs decrease their cell volumes successively with each NB division before they exit the cell cycle and differentiate (*Homem et al., 2014*; *Maurange et al., 2008*). However, tNBs do not shrink during metamorphosis (*Homem et al., 2014*) and continue to proliferate even in the adult fly brain (*Bello et al., 2006*; *Loop et al., 2004*; *Mukherjee et al., 2016*; *Narbonne-Reveau et al., 2016*). Moreover, in contrast to wild-type brains, the resulting tumor brains can be serially transplanted into host flies for years (*Caussinus and Gonzalez, 2005*; *Gateff, 1978*), indicating the immortality of these tumors.

Similarly, mammalian homologues of *numb* (*Cicalese et al., 2009*; *Colaluca et al., 2008*; *Ito et al., 2010*; *Pece et al., 2004*) and *brat* (*Chen et al., 2014*; *Mukherjee et al., 2016*) have been shown to inhibit tumor growth. Furthermore, the human *brat* homologue *TRIM3* is depleted in 24% of gliomas (*Boulay et al., 2009*; *Chen et al., 2014*) and NUMB protein levels are markedly reduced in 55% of breast tumor cases (*Pece et al., 2004*). Therefore, results obtained in these *Drosophila* tumor models are highly relevant.

Here, we used the *Drosophila brat* tumor model to investigate how tNBs differ from their physiological counterparts, the NBIIs. Our results indicate that progression towards a malignant state is an intrinsic process in *brat* tNBs that does not correlate with stepwise acquisition of DNA alterations. Transcriptome profiling of larval NBs identified the previously uncharacterized long non-coding (lnc) RNA cherub as crucial for tumorigenesis, but largely dispensable for NB development. Our data show that cherub is the first identified lncRNA to be asymmetrically segregated during mitosis into INPs, where the initial high cherub levels decrease with time. Upon the loss of *brat*, the smaller cherub-high cell reverts into an ectopic tNBs resulting in tumors with high cortical cherub. Molecularly, cherub facilitates the binding between the RNA-binding protein Staufen and the late temporal identity factor Syp and consequently tethers Syp to the plasma membrane. Depleting *cherub* in *brat* tNBs leads to the release of Syp from the cortex into the cytoplasm and represses tumor growth. Our data provide insight into how defects in asymmetric cell division can contribute to the acquisition of tumorigenic traits without the need of DNA alterations.

## Results

### Intrinsic mechanisms render tumor neuroblasts immortal

Transplanted *brat* tumors have the ability to indefinitely grow in wild-type host flies in contrast to injected control brains (*Caussinus and Gonzalez, 2005*). However, it is unclear whether the proliferation potential of tNBs is an intrinsic feature or an altered response to signals from the brain niche. For this purpose, we transplanted NBs, isolated from other brain cells by Fluorescence-activated cell sorting (FACS), into adult host flies. This procedure also allows the transplantation of equivalent NB numbers for each condition in contrast to the different NB quantities of wild-type compared to *brat* depleted brain pieces. Similar to the transplantation of tissue fragments, only *brat* [RNAi] tNBs formed tumors in host flies, whereas control NBs did not (*Figure 1C*). Thus, *brat* [RNAi] tNBs require intrinsic mechanisms that render them immortal.

## Whole genome sequencing reveals no recurrent DNA alterations in tumor neuroblasts

It is generally assumed that tumorigenesis involves multiple DNA mutations affecting proliferation control pathways (*Hanahan and Weinberg, 2011*). Nonetheless, the events leading to malignant transformation in *brat* mutants could be genetic or (post)transcriptional. To distinguish between these possibilities, we analyzed the DNA content of cells from tumor and control brains by Hoechst staining and subsequent FACS analyses. In larval and adult stages, tumor brains, similar to control brains, showed two peaks corresponding to diploid (G0/1 phase) and tetraploid (G2/M phase) karyotypes (*Figure 2A*). Although *brat* tumor cells showed an increase in tetraploid cells due to more dividing cells, we did not observe any aneuploid and polyploid karyotypes (*Figure 2A*).

To investigate potential tumor-specific DNA alterations in more detail, we sequenced the genome of three adult brain tumors from hypomorphic *brat* $^{k06028}$ mutants and compared it to that of tumor-free abdominal tissue from the same flies. To identify large genomic aberrations we compared the coverage of tumor versus control samples across the genome. As exemplified for the chromosome arm 2L (*Figure 2B*), several regions show differential coverage. However, all of these correspond to regions of the *Drosophila* genome that are known to be either amplified in follicle cells (*Claycomb and Orr-Weaver, 2005*) or under-replicated in adult polytene cells (*Nordman et al., 2011*), which are present in the abdomen but not in the brain. Thus, these copy number variations are not tumor-specific, but rather due to comparing different tissues (*Figure 2B, C*).

Besides that, tumor brains showed no recurrent amplifications or depletions of large chromosome regions, although the detection of afore mentioned biological phenomena shows the ability to detect gross chromosomal aberrations. Similarly, the few detected somatic point mutations did not overlap between the three tumor samples (*Figure 2D*). Small insertions and deletions (InDels) were mainly restricted to introns or intergenic regions (*Figure 2E*) and we did not identify regions (length 10000 bp) with such InDels common to all three tumor samples. For example, the two InDels affecting exons were identified in separate tumor samples and did not affect the same gene (*Figure 2F*). Our data of primary *brat*$^{k06028}$ tumors did not reveal recurrent DNA alterations among several tumors nor could we detect abnormal karyotypes. These results suggest that malignant transformation is unlikely to be regulated through a stepwise acquisition of DNA mutations. Thus, the cell-autonomous events leading to highly proliferative tNBs seem to be of transcriptional or post-transcriptional nature.

## The lncRNA cherub is upregulated in tumor neuroblasts compared to type II neuroblasts

To identify genes involved in brain tumorigenesis, we determined the transcriptomes of FACS-sorted tNBs and NBIIs (*Figure 3A*). As larval fly brains contain only 16 NBIIs, we developed a method that combines transposase fragmentation with molecular barcodes (DigiTAG) to derive high quality transcriptome data from low amounts of input RNA (*Figure 3B*). In total, we found 1372 up- and 345 downregulated genes in tNBs compared to NBIIs (FDR 0.01, log2fold change $\leq -2$ and $\geq 2$, respectively) (*Figure 3C*). As a quality control, we successfully verified transcriptional changes of *brat* and fifteen genes with a range of different expression levels by quantitative PCR (qPCR) (*Figure 3—figure supplement 1A–C*). In contrast to previous whole brain expression datasets (*Carney et al., 2012*), using defined cell populations in our transcriptome analysis allowed us to identify tNB-specific expression changes. The upregulated gene most highly expressed in tNB was *CR43283* (*Figure 3C*), which encodes three alternatively spliced polyadenylated transcripts without any long open reading frames. Peptides of short open reading frames found in CR43283 were not detected in the *brat* tumor proteome (*Jüschke et al., 2013*) (data not shown) and PhyloCSF analysis of *CR43283* did not reveal any coding potential (*Figure 3D*). Thus, *CR43283* likely acts as a lncRNA. To indicate its *brat* antagonizing activity (see below), we renamed *CR43283* into *cherub*, the antonym for *brat*.

## cherub is required for *brat* tumor growth

To analyze *cherub's* function, we inserted FRT sites at the 5' and 3' ends of the longest predicted transcript and generated a deletion by Flp recombination (*cherub* $^{DEL}$) (*Figure 4—figure*

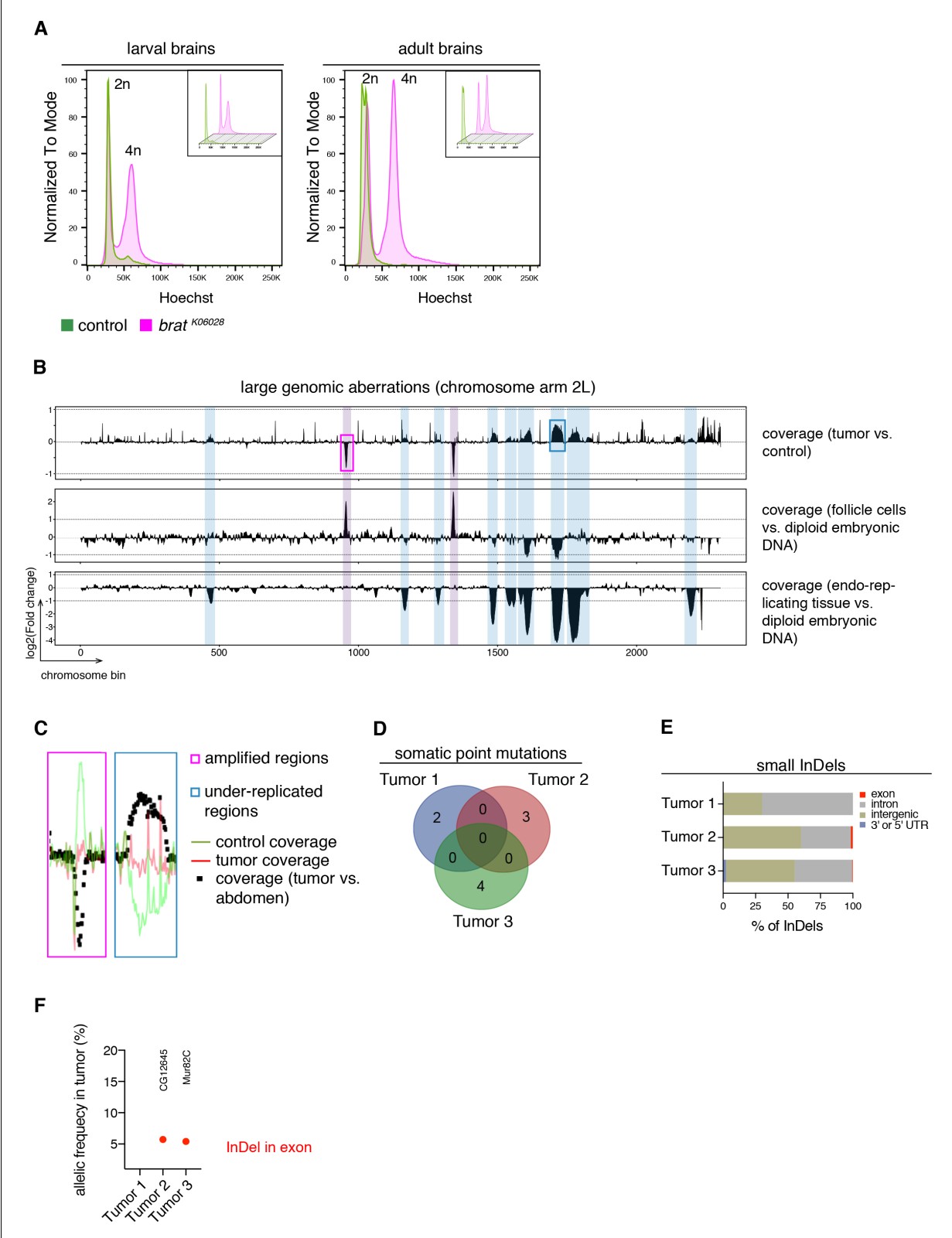

**Figure 2.** *brat* tumors form independently of common DNA alterations. (**A**) DNA content analysis of *brat* [k06028] mutant and control brains showing non-dividing (2n) and mitotic cells (4n). DNA content for each genotype is shown separately in small boxes. (**B–F**) Genome sequencing of *brat*[k06028] brains and abdominal tissue (control). Note that *Drosophila* brain cells are diploid while the control samples harbor endo-replicating tissue and follicle cells with increased DNA copy number. (**B**) A representative coverage plot of chromosome arm 2L of one tumor is shown (top row), together with datasets
*Figure 2 continued on next page*

*Figure 2 continued*

indicating under-replicated (blue) and amplified (magenta) genomic regions. (C) Close-ups of boxes marked in (B) showing steady tumor coverage (red) and a drop (under-replication) or increase (amplification) in the coverage of the control sample (green). (D) Genes with identified somatic point mutations (coverage tumor >14 and control >8, allelic frequency ≥0.1) of three tumors do not overlap. (E) Affected genomic regions of identified small InDels. (F) In tumor 2 and 3, one small InDel each was identified in an exon with low allelic frequency. Affected genes are indicated.
DOI: https://doi.org/10.7554/eLife.31347.004

supplement 1A–C). Our phenotypic analysis also included a promoter deletion created using CRISPR-Cas9 (*cherub* ^promDEL^) and an RNAi construct (*Figure 4—figure supplement 1D*) to exclude artefacts from deleting DNA elements. *cherub* mutants were viable and fertile, as numbers of eggs laid per female and of emerging adults were similar to control flies (*Figure 4A,B*). Furthermore, *cherub* mutant flies showed normal behavior in a geotaxis assay (*Figure 4C*). NBII lineages lacking *cherub* appeared normal (*Figure 4D*) and contained one large NBII expressing Deadpan (Dpn) but not Asense (Ase) surrounded by Dpn⁻Ase⁻ immature and Dpn⁺Ase⁺ mature INPs (*Figure 4E,F*). Nevertheless, in a *cherub* mutant background neither *brat* mutations nor *brat* ^RNAi^ resulted in the formation of large brain tumors (*Figure 5A,B*, *Figure 5—figure supplement 1A*). Both, tumor volumes and numbers of Dpn⁺ tNBs were strongly reduced (*Figure 5—figure supplement 1B*). The reduction in tumor growth resulted in increased viability (*Figure 5C*, *Figure 5—figure supplement 1C*). A genomic rescue construct was able to again increase the number of Dpn⁺ tNBs in *cherub brat* depleted brains (*Figure 5—figure supplement 1D*). Thus, cherub is required for tNBs to form large brain tumors.

## cherub is unequally distributed during neuroblast mitosis through binding to the RNA-binding protein Staufen

Since the subcellular localization of lncRNAs is often associated with their function (*Chen, 2016*), we visualized the expression of *cherub* in NBII lineages using a single-molecule FISH technique. *cherub* was expressed in all central brain and ventral nerve cord NB lineages (*Figure 6—figure supplement 1A*).

In NBIIs, FISH probes against cherub stained two nuclear dots (*Figure 6A*) representing nascent transcripts (*Levesque and Raj, 2013*). In addition, the lncRNA was enriched at the cell cortex and was only weakly detected in the cytoplasm (*Figure 6A*). The highest levels of *cherub* were found in newborn INPs, where it was cytoplasmic (*Figure 6A,B*). Over time, *cherub* expression in INPs decreased and the lncRNA was only weakly detected in GMCs and neurons (*Figure 6B*). The majority of NBs cease their proliferation in early pupal stages and terminally differentiate (*Homem et al., 2015*). Consequently, adult brains showed no expression of *cherub* (*Figure 6—figure supplement 1B*). Notably, this expression pattern and the localization of cherub was conserved in other *Drosophila* species (*Figure 6—figure supplement 1C,D*).

Remarkably, the cortical pool of cherub localized asymmetrically in mitotic NBs, opposite to the apical marker aPKC and thus segregated into the INP upon cytokinesis (*Figure 6C*). To avoid false positive cherub crescents due to surrounding cherub high daughter cells, we additionally confirmed the basal localization of cherub in isolated NBIIs in vitro (*Figure 6D,E*, *Video 1*).

Among all three *cherub* isoforms, only isoform RC can be unambiguously detected. However, the combination of FISH probes against specifically RC and regions present in RA and RC or RB and RC confirmed that all isoforms showed the same localization pattern and were asymmetrically segregated in NBIIs during mitosis (*Figure 6—figure supplement 1E–K*).

Previous work has shown that during mitosis the apical aPKC-Par complex releases Lethal (2) giant larvae (Lgl) from the apical side of the cell, which then promotes the localization of factors to the basal cell pole (*Betschinger et al., 2003*; *Lee et al., 2006c*). To determine whether this mechanism is involved in cherub polarization during cell division, components of the apical as well as of the basal domain were misexpressed or knocked down by RNAi. While expressing a constitutively active form of Lgl (Lgl3A) led to a uniform cortical localization of cherub, the overexpression of a constitutively active form of aPKC (aPKCΔN) resulted in cytoplasmic cherub (*Figure 6F*). Additionally, cherub transcripts became enriched in the nearest daughter cells of the NBII in control brains, but upon aPKCΔN expression cherub concentrations were comparable between both cells (*Figure 6G*),

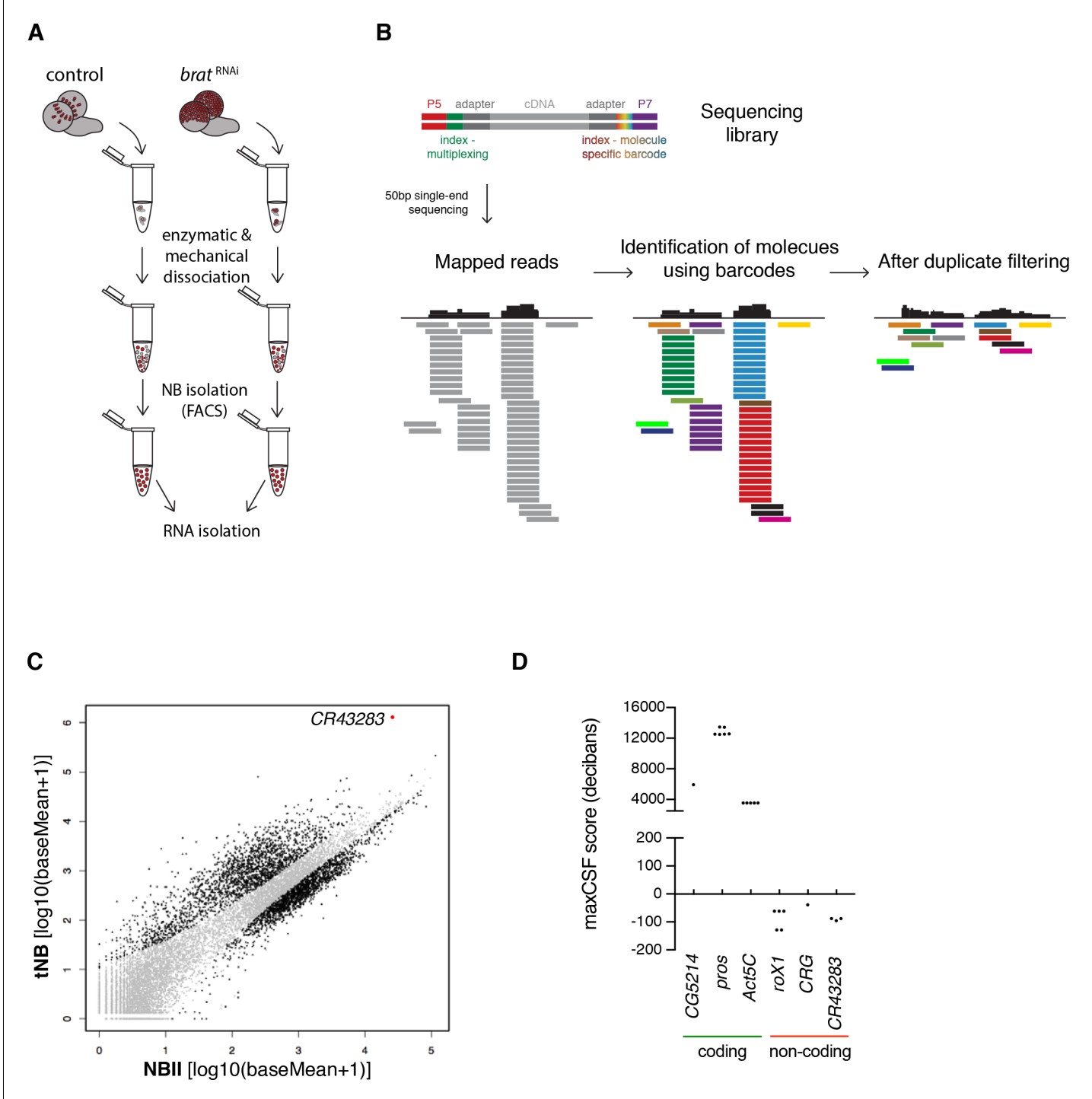

**Figure 3.** The lncRNA *CR43283* is upregulated in *brat* tumor neuroblasts. (**A**) Cartoon illustrating the strategy to isolate NBII and tNBs. (**B**) Schematic overview of DigiTAG. cDNA fragments of the sequencing library harbor random 8-mer index tags (barcodes) unique for each individual molecule. After sequencing, reads are mapped and barcodes are assigned to each read. Reads with identical barcodes are removed to avoid amplification biases. (**C**) Scatter plot showing the expression in tNBs and NBIIs of genes detected by DigiTAG. Genes significantly up- or downregulated are shown in black (FDR 0.01, p<0.01), genes unchanged in expression levels in grey. *CR43283* (*cherub*) is highlighted in red. (**D**) Protein coding genes show positive CSF scores, *CR43283* has a negative CSF score (non-coding) similar to well known lncRNAs (roX1, CRG). CSF score for each isoform of a gene is depicted.
DOI: https://doi.org/10.7554/eLife.31347.005

The following figure supplement is available for figure 3:

**Figure supplement 1.** Confirmation of up- and downregulated genes in *brat* tumor neuroblasts identified by DigiTAG.

*Figure 3 continued on next page*

*Figure 3 continued*

DOI: https://doi.org/10.7554/eLife.31347.006

indicating symmetric distribution of cherub. From these results, we conclude that the canonical asymmetric cell division machinery establishes the unequal partitioning of cherub between daughter cells.

Basal cherub polarity depended on Miranda (*Figure 6F,G*), a known adaptor protein required to tether proteins to the basal part of a NB's plasma membrane. As Miranda does not possess any RNA-binding domains, one of the proteins transported by Miranda might localize cherub. The Miranda-associated RNA-binding protein Staufen (*Matsuzaki et al., 1998*; *Schuldt et al., 1998*; *Shen et al., 1998*; *Slack et al., 2007*) has been described to enrich at the basal cell pole of dividing embryonic and larval NBs (*Broadus et al., 1998*; *Jia et al., 2015*; *Li et al., 1997*) and indeed it also localized asymmetrically in dividing NBIIs (*Figure 6—figure supplement 2A*). Consequently, Staufen was enriched in the most recently born iINPs and declined upon INP differentiation (*Figure 6—figure supplement 2B*). In *staufen* depleted NBIIs, cherub was no longer cortical, failed to segregate asymmetrically and accumulated in the cytoplasm (*Figure 6H,I*, *Figure 6—figure supplement 2C*). qPCR on RNA, isolated from anti-Staufen immunoprecipitates, detected *cherub* but not the highly expressed control RNA *RpL32* (*Figure 6J*). In-silico analysis of cherub transcripts showed limited sequence similarity across *Drosophila* species (*Figure 6—figure supplement 3A,B*), but revealed four thermodynamically stable and evolutionary conserved secondary RNA structures (*Figure 6—figure supplement 3A,C*). The identified structures resemble stem loops previously described for Staufen binding (*Ferrandon et al., 1994*; *Laver et al., 2013*) and hence may constitute potential binding sites. Notably, in some species base-pairing nucleotides were mutated in such a way to still allow base-pairing (*Figure 6—figure supplement 3C*). This indicates that the structure rather than the sequence is preserved, which is in accordance with the fact that Staufen binds double-stranded structures and not a specific sequence motif (*Ramos et al., 2000*). Therefore, binding to Staufen is required for segregating cherub into INPs to prevent its accumulation in NBIIs over time (*Figure 6K*).

## *cherub* accumulates in *brat* tumor neuroblasts

To understand how high levels of cherub emerge in *brat* tumors, we utilized the ability to detect both, nascent nuclear and cortical cherub by FISH. Intensity measurements of nuclear cherub dots, which represent transcribed RNA (*Figure 7—figure supplement 1A*), did not reveal any significant increase in *brat* [RNAi] tNBs (*Figure 7A*). This suggests that high levels are not due to transcriptional upregulation. Cortical cherub, in contrast, was strongly enhanced in *brat* tNBs (*Figure 7B*, *Figure 7—figure supplement 1B*) and still asymmetrically inherited by one daughter cell in the majority of tNBs (*Figure 7C,D*).

To understand how cherub [high] tNBs arise, we used a temperature sensitive system to induce *brat* [RNAi] for a defined time to follow cherub accumulation upon *brat* depletion over time. After 24 hr, *brat* [RNAi] NBIIs form iINPs that do not differentiate into mINPs and therefore do not re-enter the cell cycle (*Bowman et al., 2008*). In these *brat* depleted INPs, cherub remained highly expressed and cortical, whereas it became cytoplasmic in control INPs (*Figure 7E*). After 48 hr, *brat* [RNAi] iINPs revert into supernumerary Dpn[+], Ase[-] NBIIs and initiate tumor formation. Whereas control INPs started to downregulate cherub, these extra tNBs had strongly elevated cortical cherub (*Figure 7F, G*), presumably because the lncRNA was not released into a differentiating cell in which cherub levels gradually decrease. Hence, cherub accumulates in tNBs, because these supernumerary NBs arise from the daughter cells that carry high levels of cherub.

## Tumor neuroblasts show defects in temporal neuroblast identity and require early temporal factors

Over time, *Drosophila* NBs express distinct sets of genes, which confer distinct temporal identities. A temporal fate allows NBs to generate neurons with different axonal projection patterns during development and is important for their timely cell cycle exit (*Homem et al., 2014*; *Kohwi and Doe, 2013*; *Liu et al., 2015*; *Maurange et al., 2008*). Moreover, the molecular NB clock has been shown

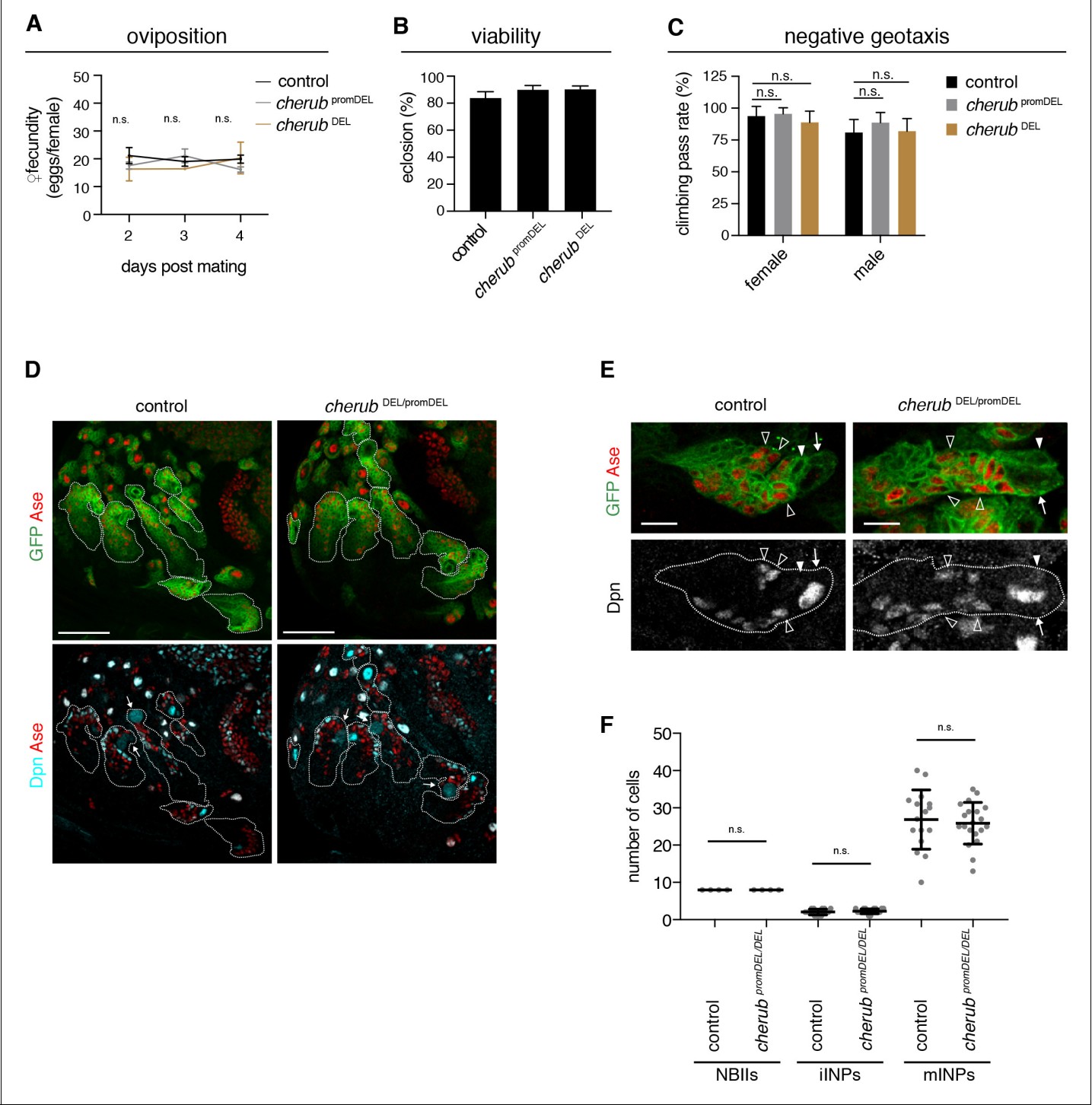

**Figure 4.** *cherub* is dispensable for brain development. (**A**) Assessment of female fecundity. For each genotype n = 3 independent crosses. Data are mean ±SD. One-way ANOVA. Not significant (n.s.). (**B**) Fertility and viability of *cherub* mutants shown as percentage of collected eggs giving rise to adult flies. Per genotype n = 10 replicates each consisting of 100 collected eggs. Data are mean ±SD. (**C**) Quantification of the negative geotaxis behavior in *cherub* mutants and control. For each genotype n = 10 replicates, each consisting of 10 adult flies. Data are mean ±SD. One-way ANOVA, not significant (n.s.). (**D**) NBII lineages (outline) of one brain lobe. NBII marked by Dpn. Dividing NBII show cytoplasmic Dpn staining (arrows). Scale bars 50 µm. (**E**) Close-ups of control and *cherub* mutant NBII lineages (outlined) show one single big Dpn+ NBII (arrow), Dpn- Ase- iINP (arrowhead) and Dpn+ Ase+ mINPs (open arrowheads). Scale bars 10 µm. (**D, E**) UAS-*dcr2; insc*-GAL4, UAS-CD8::GFP was used to outline NB lineages. (**F**) Quantifications of large Dpn+ NBIIs (n = 4 brain lobes from four different brains), Ase-Dpn- iINPs and Ase+ Dpn+ mINPs in control (n = 16 lineages from four brain lobes) and *cherub* mutant (n = 20 lineages from four brain lobes) L3 brains. Error bars show mean ±SD. Student's *t*-test. n.s. not significant.

*Figure 4 continued on next page*

*Figure 4 continued*

DOI: https://doi.org/10.7554/eLife.31347.007

The following figure supplement is available for figure 4:

**Figure supplement 1.** Analysis of *cherub* mutants.

DOI: https://doi.org/10.7554/eLife.31347.008

to be important for generating malignant tumors. When tumors are induced early in development tumor cells are generated that drive tumor proliferation even at later developmental stages due to the failure to turn-off early growth-promoting genes (*Narbonne-Reveau et al., 2016*).

Although tNBs used for transcriptome analysis were collected at late larval stages, they still expressed genes usually restricted to early NBs (24–50 hr after larval hatching) (*Liu et al., 2015*), among them the early NB identity gene *Imp* (*Figure 8A*). Temporal identity in NBIIs is controlled by opposing gradients of the RNA-binding proteins Imp and Syp (*Figure 8B*) (*Ren et al., 2017*; *Syed et al., 2017*). The transition from early Imp-positive to a late Syp-positive NB state is triggered by the nuclear receptor Seven-up (Svp) (*Narbonne-Reveau et al., 2016*; *Ren et al., 2017*; *Syed et al., 2017*).

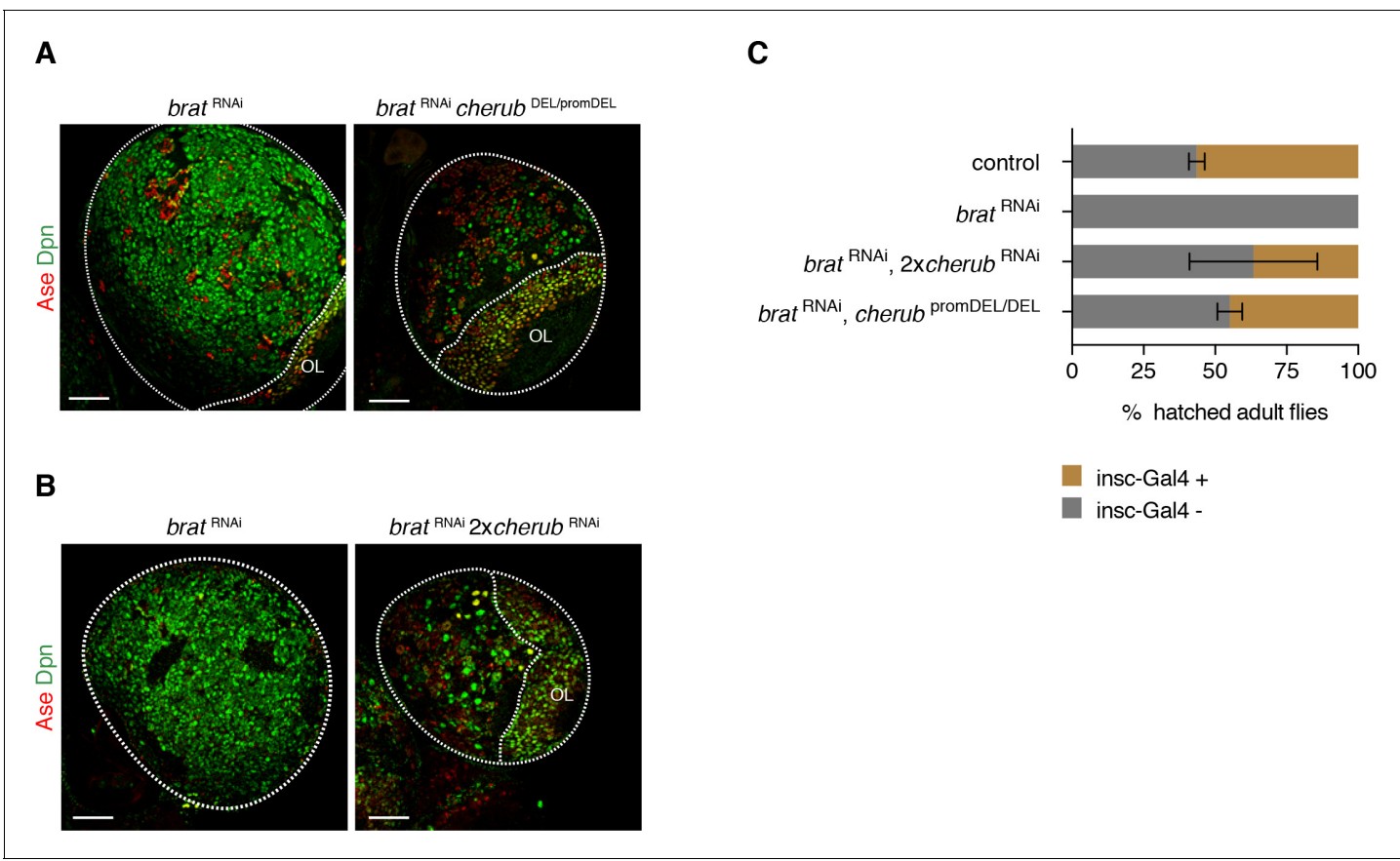

**Figure 5.** *cherub* is required for *brat* tumorigenesis. (**A, B**) Brain lobes (outlined) stained for Dpn (NB marker) and Ase (differentiating cells). OL, optic lobe. Scale bars 50 μm. Driver line was UAS-*dcr2*; *insc*-GAL4, UAS-CD8::GFP. (**C**) Percentage of hatched adult flies. *brat* [RNAi] flies were crossed to *insc*-GAL4/*CyO*. Flies with *brat* [RNAi] that inherit the balancer *CyO* hatch, but those with *insc*-GAL4 die. *cherub* mutants rescue the survival of *insc*-GAL4 + *brat* [RNAi] expressing flies. n = 2 independent experiments per genotype. For each genotype per experiment ≥50 flies were counted. Data are mean ±SD.

DOI: https://doi.org/10.7554/eLife.31347.009

The following figure supplement is available for figure 5:

**Figure supplement 1.** cherub reduces tumor growth in *brat* mutants.

DOI: https://doi.org/10.7554/eLife.31347.010

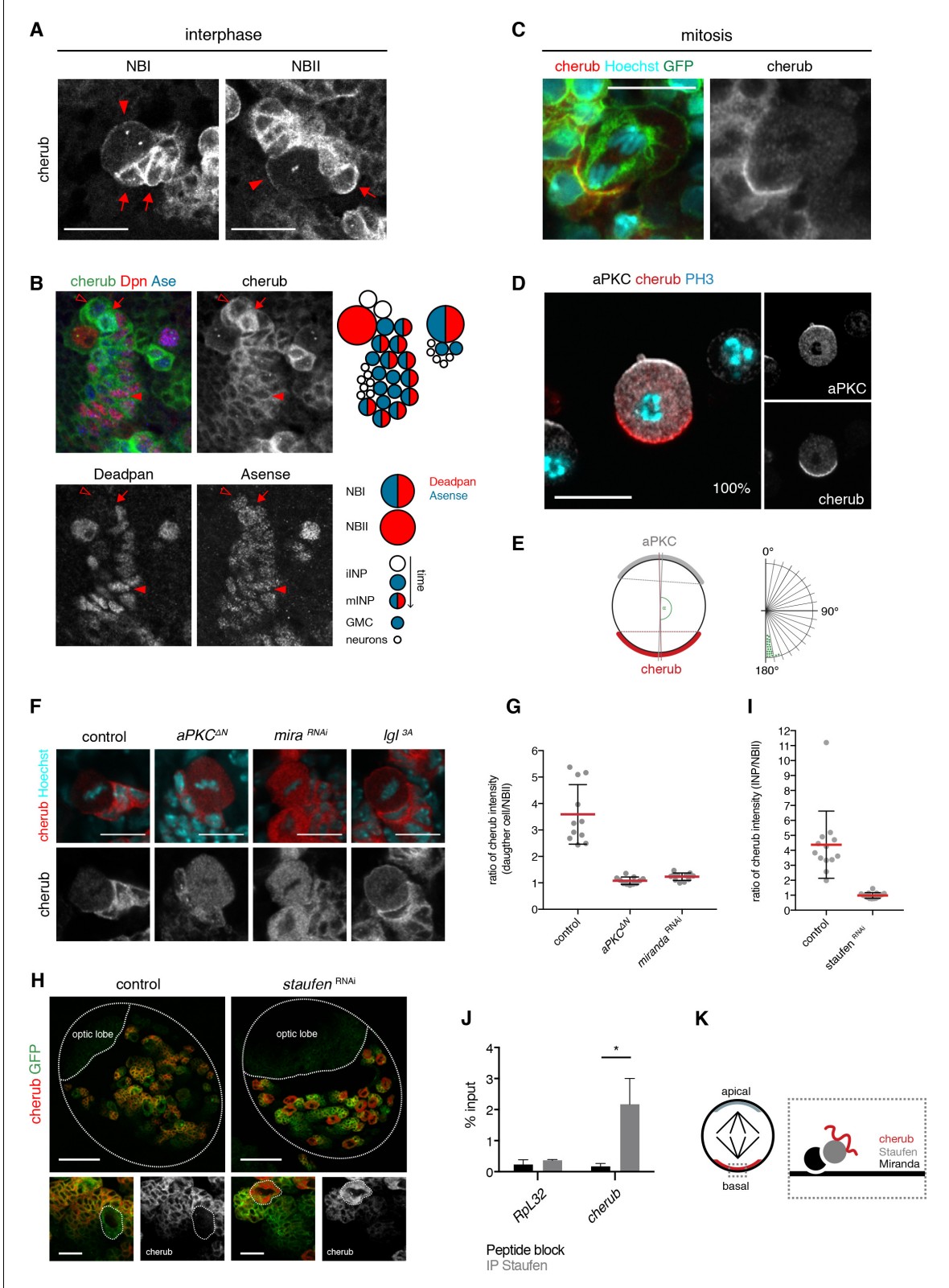

**Figure 6.** The RNA-binding protein Staufen segregates cherub asymmetrically into the INP upon cell division. (**A**) Close-up images of NB clones in interphase. In NBs (arrowhead) cherub FISH signal is detected as two dots and cortically enriched. Nearest daughter cells (arrows) show high cherub levels in the cytoplasm. Scale bars 20 μm. (**B**) Close-up of a NBI and a NBII lineage. Cell types are defined by cell size and the marker combination of Dpn and Ase according to the cartoon. Newly born Dpn⁻ iINPs (open arrowhead) have higher cherub levels than older Dpn⁺ Ase⁺ mINPs (arrowhead).

*Figure 6 continued on next page*

*Figure 6 continued*

Mitotic INPs show uniform cytoplasmic cherub localization (arrow). (**C**) Close-up images of cherub localization in a mitotic NBII outlined by membrane-bound GFP in vivo. Scale bar 10 µm. (**D**) Representative image of a dividing NBII in vitro. 100% of mitotic NBIIs show cherub crescents (**E**) opposite to the apical aPKC crescent as measured by the angle between aPKC and cherub crescents. An angle of 180° corresponds to crescents opposite each other. n = 23 NBIIs. Scale bar 10 µm. (**F**) Expressing constitutively active Lgl (Lgl$^{3A}$) leads to a uniformly cortical distribution while constitutively active aPKC (aPKC$^{\Delta N}$) or *miranda (mira)* knockdown displaces cherub from the cell cortex. Scale bars 10 µm. (**G**) Ratios of cytoplasmic cherub intensity of INP (or daughter cell) and NBII are depicted. In control, asymmetric cell division leads to higher cherub levels in INPs (n = 11 NB-INP pairs from four different brains). Upon the expression of aPKC$^{\Delta N}$ (n = 11 NB-daughter cell pairs from four different brains) or *miranda*$^{RNAi}$ (n = 12 NB-daughter cell pairs from four different brains), cherub levels are similar in NBIIs and their recently born (closest) daughter cell. Data are mean ±SD. (**H**) cherub localization in control and upon *staufen* knockdown. Brain lobes (top) and NBII (bottom) are outlined. Scale bars 50 and 10 µm. Driver line was UAS-dcr2; insc-GAL4, UAS-CD8::GFP. (**I**) In contrast to control (n = 13 NBII-INP pairs of 4 different brains), *staufen* depletion leads to similar cytoplasmic cherub levels between INP and NBII (n = 15 NBII-INP pairs of 4 different brains). Data are mean ±SD. (**J**) RIP-qPCR analysis for *cherub* and the non-Staufen target *RpL32*. Note that brain lysates from *brat*$^{k06028}$ mutants were used to enrich for NBs as cherub is only cortically localized in NBs. Data are mean ±SD. n = 3 independent RIP-qPCR experiments, Student's *t*-test. *p<0.05. (**K**) Cartoon of a mitotic NB with apical and basal crescents. Close-up of the basal crescent (right) shows that the Staufen-Miranda complex localizes cherub to the plasma membrane.

DOI: https://doi.org/10.7554/eLife.31347.011

The following figure supplements are available for figure 6:

**Figure supplement 1.** The expression and localization of cherub in the *Drosophila* CNS.

DOI: https://doi.org/10.7554/eLife.31347.012

**Figure supplement 2.** Staufen, similar to cherub, is asymmetrically segregated from the NBII into INPs.

DOI: https://doi.org/10.7554/eLife.31347.013

**Figure supplement 3.** cherub transcripts harbor conserved secondary RNA structures.

DOI: https://doi.org/10.7554/eLife.31347.014

To investigate whether tNBs might be defective in temporal NB identity, we analyzed the temporal gene expression signature of *brat* mutant tNBs. Indeed, antibody staining for the early temporal factor Imp confirmed that *brat* tumors contain tNBs of early temporal identity (*Figure 8C*). Interestingly, the formation of tNBs with an early temporal identity was dependent on *cherub* as upon the depletion of *cherub* Imp-positive tNBs are no longer present (*Figure 8C*). On the other hand, the late temporal identity factor Syp was also present in tNBs, but rather than being mainly cytoplasmic in late NBs as previously described (*Liu et al., 2015*), Syp was enriched at the cell cortex (*Figure 8D*). *brat* tumors depleted of *cherub* still expressed Syp, however Syp was no longer enriched at the cell cortex (*Figure 8D*). Thus, *brat* mutant tNBs fail to fully progress to the late stages of temporal identity in a cherub-dependent manner.

We hypothesized that if temporal identity defects contribute to tumor proliferation, 'aging' or 'rejuvenating' tNBs should have an effect on tumor growth and survival of tumor-bearing flies.

Advancing tNBs' temporal identity by *svp* overexpression (*Figure 8—figure supplement 1A–C*) or *Imp* depletion by RNAi (*Figure 8E,G,I*) significantly reduced tumor growth and increased the median survival time (50% survival rate) of tumor-bearing flies. Conversely, shifting temporal identity of *brat*$^{RNAi}$ tNBs towards younger stages by *svp* misexpression (*Figure 8—figure supplement 1A*) or *Syp*$^{RNAi}$ (*Figure 8 - F, H, J*) significantly decreased the median survival rates indicating faster tumor growth. Thus, temporal identity defects are involved in brain tumor growth.

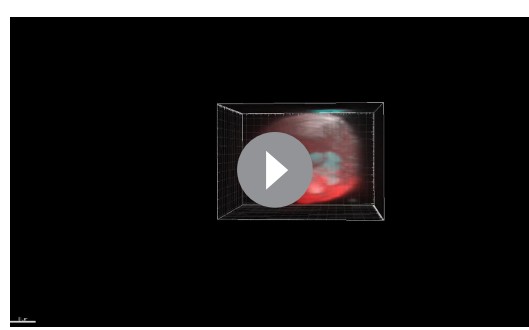

**Video 1.** cherub is restricted to the basal cell pole of mitotic type II neuroblasts. 3D reconstruction of a representative z-stack showing a NBII marked with the apical marker aPKC (white), mitotic marker PH3 (cyan) and cherub (red). First view shows the raw fluorescence signal followed by the display of reconstructed surfaces. NBIIs were harvested from dissociated brains and arrested in mitosis using Colcemid.

DOI: https://doi.org/10.7554/eLife.31347.015

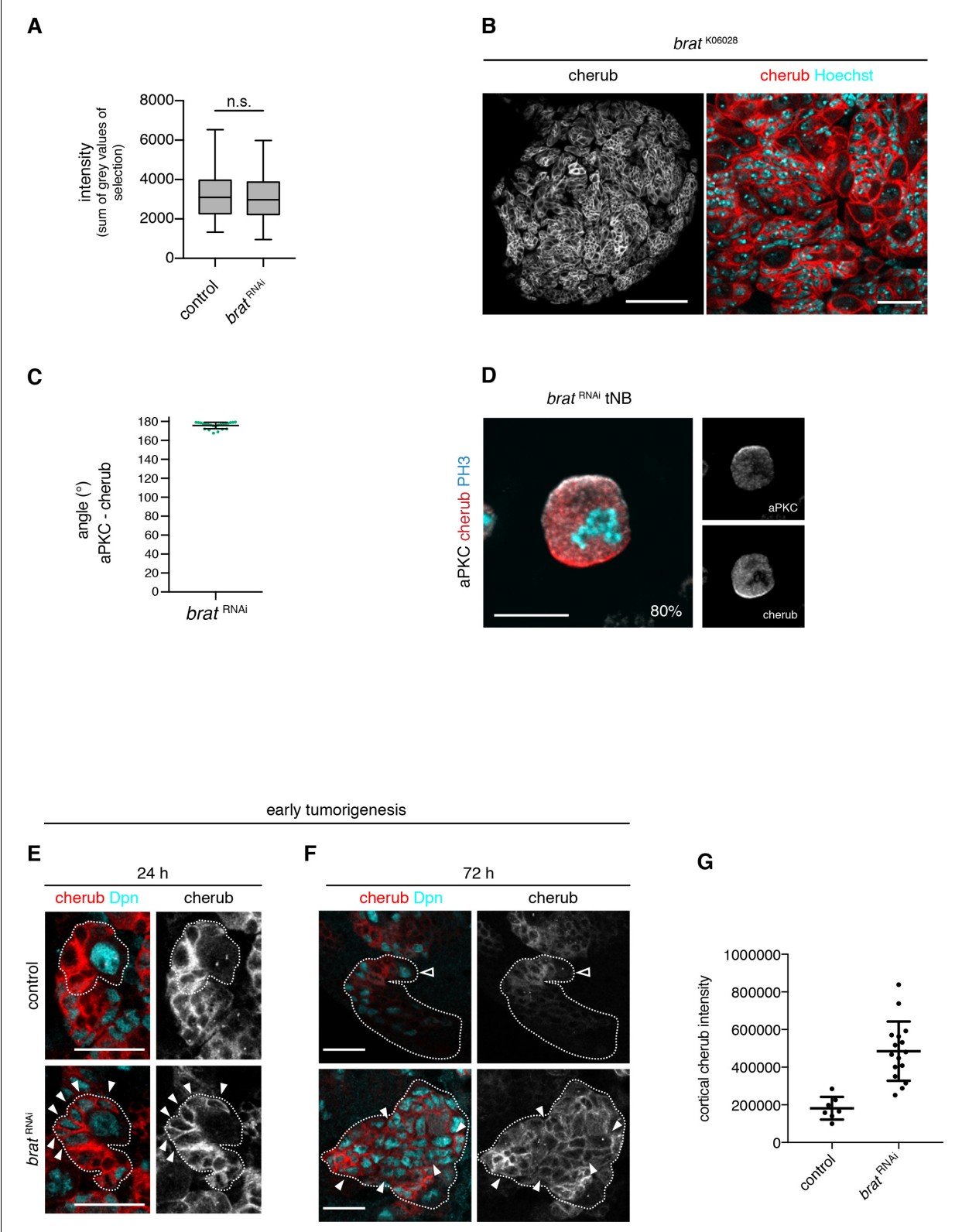

**Figure 7.** The asymmetrically segregated lncRNA *cherub* accumulates in *brat* tumor neuroblasts. (**A**) Intensity measurements of nascent cherub transcript in NBIIs (n = 43 NBs) and *brat* tNBs (n = 43 tNBs) from three independent FISH experiments. Student's *t*-test. Not significant (n.s.) p>0.05. (**B**) cherub localization in a *brat* mutant brain lobe (left, scale bar 100 µm) and close-up of tNBs (right, scale bar 20 µm). (**C**) Quantification of the angle between aPKC and cherub crescents of mitotic tNBs in vitro. An angle of 180° corresponds to a basal cherub crescent. n = 23 mitotic tNBs. Mean ±SD

*Figure 7 continued on next page*

*Figure 7 continued*

is shown. (D) Representative image of a mitotic tNB in vitro. Scale bar 10 μm. Percentage of tNBs showing a cherub crescent during cell division is indicated. n = 82 mitotic tNBs. (E, F) *cherub* expression in NBII clones (outlined) induced with the TARGET system for 24 hr (E) or 72 hr (F). Arrowheads exemplify Dpn[+] tNBs and the open arrowhead marks the NBII. Scale bars 20 μm. UAS transgenes are expressed by UAS-*dcr2; wor*-GAL4, *ase*-GAL80; UAS-CD8::GFP. (G) Quantification of cortical cherub intensity after 72 hr clone induction. Control n = 7 NBIIs and *brat* [RNAi]n = 16 tNBs. Data are mean ±SD. Student's t-test and ****p<0.0001.

DOI: https://doi.org/10.7554/eLife.31347.016

The following figure supplement is available for figure 7:

**Figure supplement 1.** Cortical cherub levels are increased in *brat* tNBs.

DOI: https://doi.org/10.7554/eLife.31347.017

## cherub facilitates the interaction between Staufen and late temporal identity factor Syncrip

Since the cortical localization of Syp is cherub-dependent in tNBs, it is tempting to speculate that cherub acts as an adapter to facilitate the binding between Staufen and Syp. This assumption implies that firstly Syp shows a similar localization pattern to Staufen and cherub and secondly Syp segregates asymmetrically via Staufen into iINPs. Indeed, in NBIIs Syp was localized at the cortex and also detected in the cytoplasm (*Figure 9D,E*). Similar to Staufen and cherub, Syp segregated asymmetrically in mitosis (*Figure 9A–C*), and became enriched in the cytoplasm of the most recently-born INPs (*Figure 9D,E*). Furthermore, Syp colocalized with Staufen and cherub in mitotic NBIIs (*Figure 9—figure supplement 1A–G*). In *cherub* mutants or upon *staufen* knockdown, Syp was exclusively cytoplasmic and no longer cortical (*Figure 9D,E*). These results indicate that Syp is recruited to the cortex by the cherub-Staufen complex.

To further confirm the role of cherub in Staufen-Syp complex formation, we performed immunoprecipitation experiments on brain lysates from *brat* tumors. Co-immunoprecipitation experiments demonstrated that Staufen and Syp are in a complex and binding between the two proteins was lost upon RNA digestion or *cherub* depletion (*Figure 9F,G*). RIP-qPCR experiments further confirmed that cherub was enriched upon HA-tagged Syp pull-down (*Figure 9H*). These data suggest that cherub functions as adapter between Syp and Staufen (*Figure 9I*).

## Discussion

Tumor cells display features not normally found in healthy cells. How those features arise and how they can be exploited to combat tumorigenesis are important questions in tumor biology. Here we show that the lncRNA *cherub* regulates a critical step in *Drosophila* brain tumorigenesis, but is dispensable for normal development. During each division, cherub is depleted from the NB and is segregated into the differentiating daughter cells by binding to the RNA-binding protein Staufen. During tumor formation, however, de-differentiation of the cherub-high cells results in the formation of tumor NBs with extraordinarily high cherub levels. We show that cherub induces the formation of a cortical complex between Staufen and Syp, a conserved RNA-binding protein (*McDermott et al., 2012*) required for NBs to acquire late temporal identity (*Liu et al., 2015*; *Ren et al., 2017*; *Syed et al., 2017*). Our data suggest that temporal identity defects contribute to tumor formation and reveal a new function for lncRNAs in controlling the subcellular localization of protein determinants.

## The malignant transformation of *brat* tumor neuroblasts is regulated by epigenetic mechanisms

It is commonly assumed that cancer cells become malignant and gain replicative immortality by acquiring genetic lesions (*Hanahan and Weinberg, 2011*). Surprisingly, however, our data indicate that *brat* tumors do not require additional genetic lesions for the transition to an immortal state. This is not a general feature of *Drosophila* tumors as genomic instability alone can induce tumors in *Drosophila* epithelial cells (*Dekanty et al., 2012*) and intestinal stem cells (*Siudeja et al., 2015*). However, our results are supported by previous experiments demonstrating that defects in genome integrity do not contribute to primary tumor formation in NBs (*Castellanos et al., 2008*). Similarly, tumors induced by loss of epigenetic regulators in *Drosophila* wing discs do not display genome

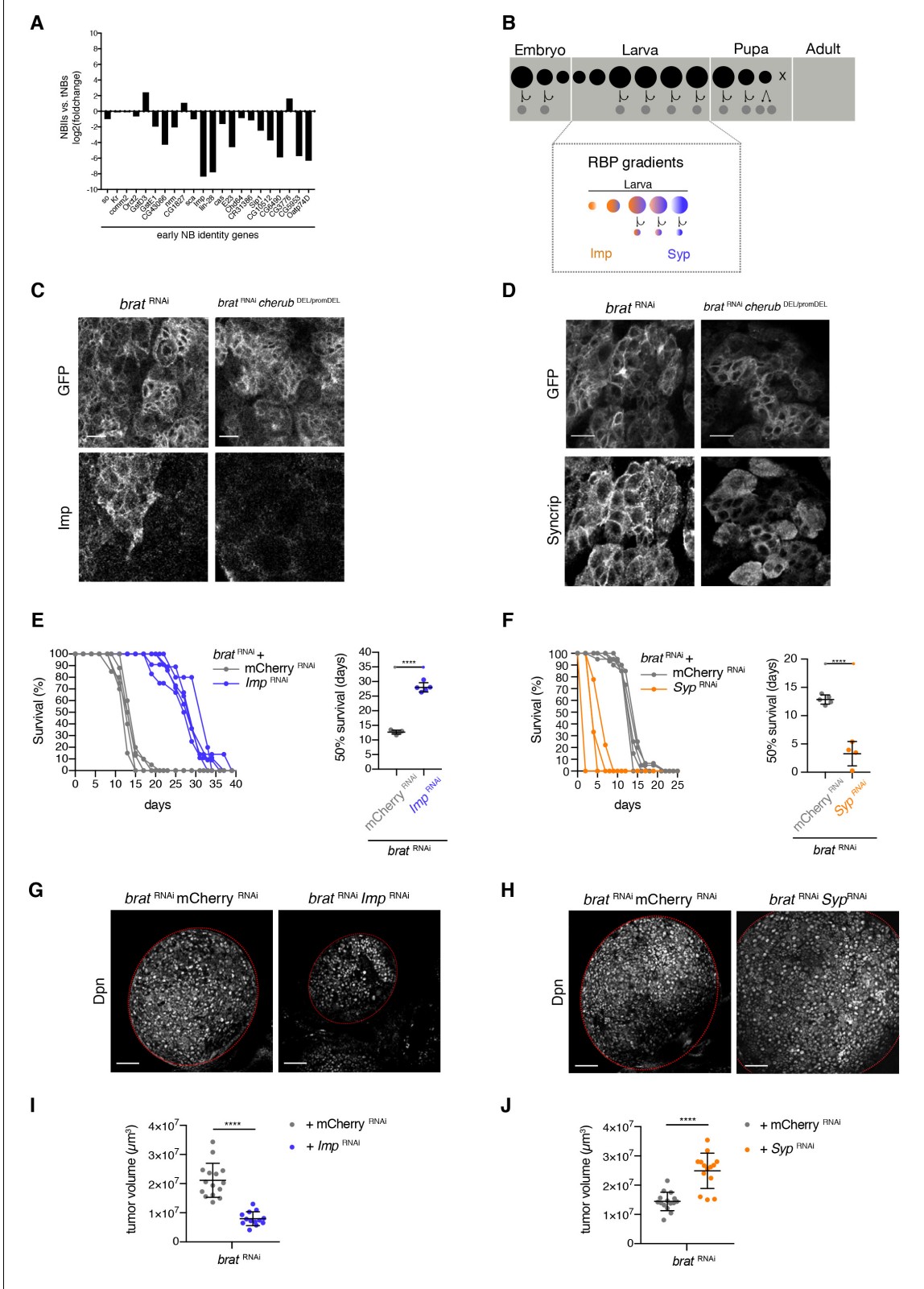

**Figure 8.** Temporal neuroblast identity controls *brat* tumor growth. (**A**) The expression of genes characteristic for a young NB identity are upregulated in tNBs compared to control NBIIs. Depicted is the log2 foldchange in expression of NBIIs versus tNBs. (**B**) Cartoon depicting the control of larval temporal NB identity by opposing RNA-binding protein (RBP) gradients (**C**) Expression of the early NB identity gene *Imp* in *brat* [RNAi] and *brat* [RNAi]*cherub* [DEL/promDEL] tNBs. (**D**) Syncrip localization in *brat* [RNAi] and *brat* [RNAi]*cherub* [DEL/promDEL] tNBs. (**C, D**) Scale bars 10 μm. Driver line UAS-*dcr2;*

*Figure 8 continued*

*wor*-GAL4, *ase*-GAL80; UAS-CD8::GFP. Membrane-bound GFP outlines tumor tissue. (E, F) Survival rates of adult flies bearing primary control tumors or rejuvenated (orange) or aged (purple) *brat* tumors. Student's *t*-test. ****p<0.0001, n ≥ 4 independent survival experiments. 50% survival rate shown as mean ±SD. (G, H) Overview of *brat* brain lobes (outlined) stained with the NB marker Dpn upon downregulation of *Imp* (G) or *Syp* (H). Scale bars 50 μm. (I, J) Quantification of tumor volume from three independent experiments. Data are mean ±SD. Student's *t*-test. ****p<0.0001. (I) Knockdown of *Imp* (n = 15 brain lobes) results in smaller tumors compared to control tumors (n = 13 brains lobes). (J) Tumors with reduced Syp levels (n = 15 brain lobes) are larger than control tumors (n = 14 brain lobes). (E–J) Transgenes were expressed using the NBII-specific driver line UAS-*dcr2; wor*-GAL4, *ase*-GAL80; UAS-CD8::GFP.

DOI: https://doi.org/10.7554/eLife.31347.018

The following figure supplement is available for figure 8:

**Figure supplement 1.** *brat* tumor growth requires a young neuroblast identity.

DOI: https://doi.org/10.7554/eLife.31347.019

instability (*Sievers et al., 2014*). In addition, the short time it takes from the inactivation of *brat* to the formation of a fully penetrant tumor phenotype would most likely be insufficient for the acquisition of tumor-promoting DNA alterations. More likely, the enormous self-renewal capacity and fast cell cycle of *Drosophila* NBs requires only minor alterations for the adoption of malignant growth. Interestingly, epigenetic tumorigenesis was described before in humans, where childhood brain tumors only harbor an extremely low mutation rate and very few recurrent DNA alterations (*Lee et al., 2012*; *Mack et al., 2014*; *Northcott et al., 2012*; *Parsons et al., 2011*; *Pugh et al., 2012*; *Robinson et al., 2012*; *Sausen et al., 2013*; *Wu et al., 2014*; *Zhang et al., 2012*). Comparable observations have been made for leukemia (*Quesada et al., 2011*; *Yan et al., 2011*). Our results might help to understand mechanisms of epigenetic tumor formation, which are currently unclear in humans.

## cherub is the first lncRNA known to segregate asymmetrically

*cherub* is the first lncRNA described to segregate asymmetrically during mitosis. Once cherub is allocated through binding to the RNA-binding protein Staufen into the cytoplasm of INPs, its levels decrease over time. Our results show that the inability to segregate cherub into differentiating cells leads to its accumulation in tNBs. The increasing amount of tumor transcriptome data indicates that a vast number of lncRNAs show increased expression levels in various tumor types (*Huarte, 2015*). Intriguingly, the mammalian homologue of cherub's binding partner Staufen has been also described to asymmetrically localize RNA in dividing neural stem cells (*Kusek et al., 2012*; *Vessey et al., 2012*). Hence, besides transcriptional upregulation, asymmetric distribution of lncRNAs between sibling cells might play a role in the accumulation of such RNAs in mammalian tumors.

## Defects in temporal identity contribute to tumorigenic growth

Our data suggest a functional connection between cherub and proteins involved in temporal neural stem cell patterning. We show that tNBs retain the early temporal identity factor Imp even during late larval stages. However, *Imp* expression in *brat* mutants is heterogeneous and only a subpopulation of tNBs maintains young identity.

Tumor heterogeneity has also been described for *pros* tumors, where only a subset of tNBs maintains expression of the early temporal factors Imp and Chinmo (*Narbonne-Reveau et al., 2016*). Interestingly, it is this subpopulation that drives tumor growth in *prospero* tumors.

Consistent with this, our genetic experiments show that 'rejuvenating' tNBs enhances tumor growth and consequently increases the survival of tumor bearing flies, whereas 'aging' tNBs identity has the opposite effect. Although mammalian counterparts of Imp have not yet being shown to act as temporal identity genes, their upregulated expression has been implied in various cancer types (*Dai et al., 2017*; *Lederer et al., 2014*). Therefore, temporal patterning of NBs has an essential role in brain tumor propagation in *Drosophila.*

## cherub regulates Syp localization in *brat* tumors

The subset of tNBs that retain early identity in tumors is lost in a *cherub* mutant background. This suggests that cherub itself might regulate temporal identity. In NBs and tNBs cherub regulates Syp localization by facilitating the binding of Syp to Staufen and thus recruiting it to the cell cortex. In

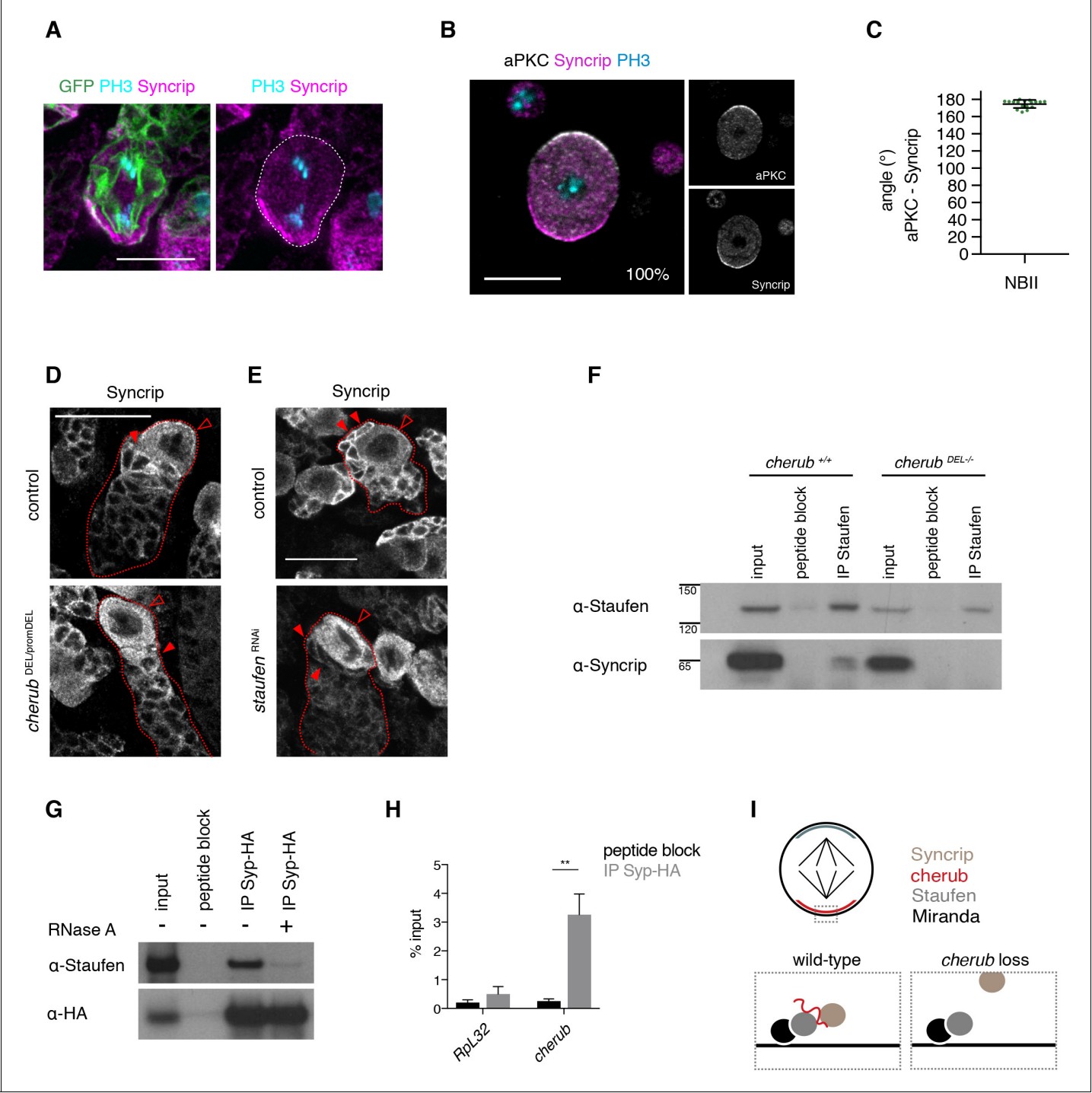

**Figure 9.** cherub facilitates the binding between Staufen and the late temporal factor Syncrip. (**A**) Syncrip (Syp) localization in a mitotic NB (outlined) in vivo. Scale bar 10 μm. (**B**) Representative image and percentages of mitotic NBIIs in vitro showing Syp crescents. n = 17 mitotic NBIIs. Scale bar 10 μm. (**C**) Angle measurement between Syp and apical aPKC crescents. n = 17 mitotic NBIIs. Mean ±SD is shown. (**D, E**) Syp is delocalized in NBIIs (open arrowhead) and cytoplasmic enrichment in newly born INPs (arrowhead) is lost in cherub mutants (**D**) or upon Staufen reduction (**E**). Scale bars 20 μm. (**E**) UAS-dcr2; insc-GAL4, UAS-CD8::GFP was used. (**F**) Immunoprecipitation of Staufen in the presence and absence of cherub. (**G**) Immunoprecipitation of expressed HA-tagged Syp in the presence and absence of RNase treatment. (**H**) RIP-qPCR analysis of cherub and the negative control RpL32 upon Syp-HA pull down. Student's t-test, \*\*p<0.01, n = 3 independent RIP-qPCR experiments, depicted as mean ±SD. (**F–H**) Brain lysates from brat [RNAi] tumors and brat [RNAi] cherub [DEL-/-] or brat [RNAi] UAS-Syp-HA were used to enrich for NBs. Driver line used was UAS-dcr2; wor-GAL4, ase-GAL80; UAS-CD8::GFP. (**I**) Cartoon summarizing the role of cherub in Miranda-Staufen-Syncrip complex formation.

DOI: https://doi.org/10.7554/eLife.31347.020

*Figure 9 continued on next page*

*Figure 9 continued*

The following figure supplement is available for figure 9:

**Figure supplement 1.** cherub, Staufen and Syncrip localize to the basal pole of mitotic NBIIs.

DOI: https://doi.org/10.7554/eLife.31347.021

tumors depleted of *cherub*, Syp localizes mainly to the cytoplasm and is no longer observed at the cortex. As the removal of Syp in tNBs leads to enhanced tumor growth and early lethality, those data suggest that cherub could control temporal NB identity by regulating the subcellular localization of Syp.

How could cherub regulate the function of Syp? The RNA-binding protein Syp is a translational regulator (*Duning et al., 2008*; *McDermott et al., 2012*; *2014*) and has been suggested to control mRNA stability (*Yang et al., 2017b*). As mammalian SYNCRIP/hnRNP Q interacts with a lncRNA that suppresses translation (*Duning et al., 2008*; *Kondrashov et al., 2005*), cherub might regulate Syp to inhibit or promote the translation of a subset of target mRNAs. In particular, in NBs Syp acts at two stages in NBs during development: Firstly, approximately 60 hr after larval hatching it represses early temporal NB factors, like Imp (*Syed et al., 2017*; *Yang et al., 2017a*). Secondly, at the end of the NB lifespan Syp promotes levels of the differentiation factor *prospero* to facilitate the NB's final cell cycle exit (*Yang et al., 2017a*; *Yang et al., 2017b*). As *cherub* depletion in *brat* tumors leads to decreased tumor growth (*Figure 5A,B*), it is possible that cherub inhibits the Syp-dependent repression of the early factor Imp, which we showed is required for optimal tumor growth (*Figure 8E,G*). However, *cherub* mutant NBIIs do not show altered timing or expression of Imp during development (data not shown). In accordance, *brat* tumors show high cortical cherub levels, but only a subset of NBs expresses Imp (*Figure 8C*). Rather than rendering Syp completely inactive, we suggest that cherub decreases the ability of Syp to promote factors important to restrict NB proliferation. As *prospero* is not expressed in NBIIs, it remains to be investigated which Syp targets are affected by cherub.

## cherub is largely dispensable for normal development

Remarkably, *cherub* mutants are viable, fertile and do not affect NBII lineages. Neurons generated by NBIIs predominantly integrate into the adult brain structure termed central complex, which is important for locomotor activity. As *cherub* mutants show normal geotaxis, function of the lncRNA seems dispensable for NBIIs to generate their neural descendants.

Nevertheless, the conserved secondary RNA structures of *cherub* and its conserved expression pattern in other *Drosophila* species suggest that it has a functional role. There are several possibilities why we do not observe a phenotype upon the loss of cherub. In wild-type flies cherub might confer robustness. A similar scenario was observed in embryonic NBs, in which Staufen segregates *prospero* mRNA into GMCs (*Broadus et al., 1998*; *Schuldt et al., 1998*). The failure to segregate *prospero* mRNA does not result in a phenotype, but it enhances the hypomorphic *prospero* GMC phenotype (*Broadus et al., 1998*). Thus segregation of *prospero* mRNA serves as support for Prospero protein to induce a GMC fate. Similarly, cherub could act as a backup to reliably establish correct Syp levels in NBIIs and in INPs. Alternatively, cherub might fine-tune the temporal patterning by regulating the cytoplasmic pool of Syp in the NBs. Increasing Syp levels have been suggested to determine distinct temporal windows, in which different INPs and ultimately neurons with various morphologies are sequentially born (*Ren et al., 2017*). Therefore, we cannot exclude that changes in Syp levels lead to subtle alterations in the number of certain neuron classes produced by NBIIs that only reveal themselves in pathological conditions like tumorigenesis.

## Future directions

Our study illustrates how a lncRNA can control the subcellular localization of temporal factors. In addition to temporal NB identity, Syp regulates synaptic transmission and maternal RNA localization (*McDermott et al., 2012*). While *cherub* is not expressed in ovaries or adult heads, Staufen has been implicated in these processes, suggesting that other RNAs might act similarly to *cherub*. Interestingly, the mammalian Syp homolog hnRNP Q binds the noncoding RNA BC200 (*Duning et al., 2008*), whose upregulation is used as a biomarker in ovarian, esophageal, breast and brain cancer

(*De Leeneer and Claes, 2015*; *Perez et al., 2008*; *Zhao et al., 2016*) (*De Leeneer and Claes, 2015*; *Perez et al., 2008*; *Zhao et al., 2016*). In the future, it will be interesting to investigate whether the mechanism we have identified in *Drosophila* is involved in mammalian tumorigenesis as well.

# Materials and methods

**Key resources table**

| Reagent type (species) or resource | Designation | Source or reference | Identifiers | Additional information |
|---|---|---|---|---|
| gene (*D. melanogaster*) | cherub | NA | FLYB:FBgn0262972 | |
| gene (*D. melanogaster*) | Syncrip | NA | FLYB:FBgn0038826 | |
| gene (*D. melanogaster*) | Staufen | NA | FLYB:FBgn0003520 | |
| gene (*D. melanogaster*) | Imp | NA | FLYB:FBgn0285926 | |
| antibody | anti-Deadpan (guinea pig, polyclonal) | PMID:24630726 | | (1:1000) |
| antibody | anti-Elav (rat, monoclonal) | Developmental Studies Hybridoma Bank | 7E8A10, RRID:AB_528218 | (1:200) |
| antibody | anti-Staufen (rat polyclonal) | PMID:19481457 | | (1:100) |
| antibody | anti-Staufen (goat polyclonal) | Santa Cruz Biotechnology | dN-16, sc-15823, RRID:AB_661413 | (1:2000) |
| antibody | anti-Asense (rat polyclonal) | PMID:24630726 | | (1:500) |
| antibody | anti-HA (mouse monoclonal) | Roche | clone 12CA5, 11583816001 | |
| antibody | anti-aPKC (rabbit polyclonal) | Santa Cruz Biotechnology | sc-216, RRID:AB_2300359 | (1:500) |
| antibody | anti-GFP (chicken polyclonal) | Abcam | ab13970, RRID:AB_300798 | (1:500) |
| antibody | anti-Miranda (rabbit polyclonal) | PMID:24630726 | | (1:500) |
| antibody | secondary antibody (HRP-linked Whole Antibodies) | GE Healthcare, Abcam, Santa Cruz Biotechnologies | sc-2384, NA931V, ab6908, RRID:AB_955425, RRID:AB_634814 | 1:2000 |
| antibody | anti-Syncrip (guinea pig polyclonal) | PMID:23213441 | | (1:500) |
| antibody | anti-PH3(Ser10) (mouse monoclonal) | Cell Signalling Technologies | 9706S, RRID:AB_331748 | (1:1000) |
| antibody | Alexa 405, 488, 568, 647 | Invitrogen | Alexa Fluor dyes | (1:1500) |
| other | Hoechst 33342 | Thermo Scientific Fisher | 62249 | (20 µM) |
| chemical compound | Colcemid | Enzo Life Sciences | ALX-430–033 M001 | (25 µM) |
| chemical compound | Protein G Dynabeads | Thermo Scientific Fisher | 10004D | |
| Recombinant DNA reagent | genomic cherub region (Bac clone) | Pacman Resources | CH322-116G10, FLYB:FBcl0758452 | |
| Recombinant DNA reagent | Walium20 vector | Plasmid Repository of DNA Resource Core(Harvard Medical School) | pWalium20 | |
| other | FISH probes labeled with Quasar Dye | Biosearch Technologies | cherub all | this study - oligonucleotide sequence see Materials and methods |
| other | FISH probes labeled with Quasar Dye | Biosearch Technologies | cherub RA/RC | this study - oligonucleotide sequence see Materials and methods |
| other | FISH probes labeled with Quasar Dye | Biosearch Technologies | cherub RB/RC | this study - oligonucleotide sequence see Materials and methods |
| other | FISH probes labeled with Quasar Dye | Biosearch Technologies | cherub RC | this study - oligonucleotide sequence see Materials and methods |

*Continued on next page*

*Continued*

| Reagent type (species) or resource | Designation | Source or reference | Identifiers | Additional information |
|---|---|---|---|---|
| software,algorithm | JACoP | PMID:17210054 | | |
| software,algorithm | Prism 7 | GraphPad Software | RRID:SCR_002798 | |
| software,algorithm | PhyloCSF | PMID:21685081 | | |
| software,algorithm | FlowJo software | FlowJo, LLC | RRID:SCR_008520 | |
| software,algorithm | BWA | PMID:19451168 | RRID:SCR_015853 | |
| software,algorithm | Picard tools (v1.82) | Broad Institute | RRID:SCR_006525 | http://broadinstitute.github.io/picard |
| software,algorithm | GATK (v2.3) | Broad Institute | RRID:SCR_001876 | https://software.broadinstitute.org/gatk/ |
| software,algorithm | MuTect (v1.1.4) | PMID:23396013 | RRID:SCR_000559 | |
| software,algorithm | SnpEff (v3.2a) | PMID:22728672 | RRID:SCR_005191 | |
| software,algorithm | Published aCGH datasets | PMID:22090375 and 21724831 | | |
| software,algorithm | TopHat | PMID:19289445 | RRID:SCR_013035 | |
| software,algorithm | DESeq (v1.10.1) | PMID:20979621 | RRID:SCR_000154 | |
| software,algorithm | HTSeq | PMID:25260700 | RRID:SCR_005514 | |
| software,algorithm | Biostrings | R package version 2.40.2. | | |
| commercial assay | Nextera DNA Library Preparation Kit | Illumina | FC-121–1031 | |
| commercial assay | Agencourt AMPure XP beads | Beckman Coulter | A63880 | |
| commercial assay | TRIzol LS | Ambion | 10296010 | |
| commercial assay | NEBNext Ultra DNA Library Prep Kit | Illumina | E7370S | |
| commercial assay | Agilent High Sensitivity DNA Kit | Agilent Technologies | 5067–4626 | |
| other | Rinaldini solution | PMID:22884370 | | 1X |
| Peptide | Staufen Blocking peptide | Santa Cruz Biotechnology | sc-15823P | used 10x amount of primary antibody |
| Peptide | HA Blocking Peptide | Roche | 11666975001 | used 10x amount of primary antibody |
| genetic reagent (*Drosophila melanogaster*) | UAS-*brat* [RNAi] | Vienna Drosophila Resource Center | VDRC:105054 and 31333 | |
| genetic reagent (*Drosophila melanogaster*) | UAS-*mira* [RNAi] | PMID:16564014 | | |
| genetic reagent (*Drosophila melanogaster*) | UAS-mCherry [RNAi] | Bloomington Drosophila Stock Center | BDSC:35785, RRID:BDSC_35785 | |
| genetic reagent (*Drosophila melanogaster*) | UAS-*aPKC*$^{\Delta N}$ | PMID:12629552 | | |
| genetic reagent (*Drosophila melanogaster*) | UAS-*staufen* [RNAi] | Vienna Drosophila Resource Center | VDRC:106645 | |
| genetic reagent (*Drosophila melanogaster*) | UAS-*Syp* [RNAi] | Vienna Drosophila Resource Center | VDRC:33012 | |
| genetic reagent (*Drosophila melanogaster*) | UAS-*Syp*-RB-HA | PMID:26472907 | | |
| genetic reagent (*Drosophila melanogaster*) | UAS-*Imp* [RNAi] | Bloomington Drosophila Stock Center | BDSC:34977, RRID:BDSC_34977 | |
| genetic reagent (*Drosophila melanogaster*) | UAS-*svp* [RNAi] | Vienna Drosophila Resource Center | VDRC:37087 | |
| genetic reagent (*Drosophila melanogaster*) | UAS-*svp* | gift from Y. Hiromi | | |

*Continued on next page*

*Continued*

| Reagent type (species) or resource | Designation | Source or reference | Identifiers | Additional information |
|---|---|---|---|---|
| genetic reagent (*Drosophila melanogaster*) | *brat k06028* | PMID:10949924 | | |
| genetic reagent (*Drosophila melanogaster*) | UAS-*dcr2*; *wor*-GAL4, *ase*-GAL80; UAS-CD8::GFP | PMID:21549331 | | |
| genetic reagent (*Drosophila melanogaster*) | UAS-*dcr2*; *insc*-GAL4, UAS-CD8::GFP | PMID:21549331 | | |
| genetic reagent (*Drosophila melanogaster*) | UAS-*stinger*::RFP | PMID:11056799 | | |
| genetic reagent (*Drosophila melanogaster*) | UAS-*stinger*::GFP | PMID:11056799 | | |
| genetic reagent (*Drosophila melanogaster*) | *tubulin*-GAL80ts | Bloomington Drosophila Stock Center | BDSC:7017, RRID:BDSC_7017 | |
| genetic reagent (*Drosophila simulans*) | *D. simulans* | Drosophila Species Stock Center | 14021–0251.265 | |
| genetic reagent (*Drosophila willistoni*) | *D. willistoni* | Drosophila Species Stock Center | 14030–0814.10 | |
| genetic reagent (*Drosophila melanogaster*) | UAS-*cherub* RNAi | this paper | | RNAi against cherub |
| genetic reagent (*Drosophila melanogaster*) | *cherub*DEL | this paper | | full deletion of cherub locus |
| genetic reagent (*Drosophila melanogaster*) | *cherub*promDEL | this paper | | deletion of promotor region of cherub |
| genetic reagent (*Drosophila melanogaster*) | cherubgenomic rescue | this paper | | genomic rescue construct of cherub, BAC clone CH322-116G10 |

## Fly strains and husbandry

The following *Drosophila* stocks were used: UAS-*brat* RNAi (VDRC, TID 105054 and 31333), UAS-*mira* RNAi (**Betschinger et al., 2006**), UAS-mCherry RNAi (BL35785), UAS-*lgl* 3A (**Betschinger et al., 2003**), UAS-*aPKC*ΔN (**Betschinger et al., 2003**), UAS-*staufen* RNAi (VDRC,TID106645), UAS-*Syp* RNAi (VDRC, TID33012), UAS-*Syp*-RB-HA (**Liu et al., 2015**), UAS-*Imp* RNAi (BL34977), UAS-*svp* RNAi (VDRC, TID 37087), UAS-*svp* (gift from Y. Hiromi), *brat* k06028 (**Arama et al., 2000**), UAS-*dcr2*; *wor*-GAL4, *ase*-GAL80; UAS-CD8::GFP (**Neumüller et al., 2011**), UAS-*dcr2*; *insc*-GAL4, UAS-CD8::GFP (**Neumüller et al., 2011**), UAS-*stinger*::RFP and UAS-*stinger*::GFP (**Barolo et al., 2000**), *tubulin*-GAL80ts (BL7017), *D. simulans* (Drosophila Species Stock Center, 14021–0251.265), *D. willistoni* (Drosophila Species Stock Center, 14030–0814.10). Stocks generated in this study: UAS-*cherub* RNAi, *cherub* DEL, *cherub* promDEL, cherubgenomic rescue. For the latter, the BAC construct CH322-116G10 was integrated into the attP40 landing site via integrase-mediated transgenesis.

If not stated otherwise, UAS transgenes were expressed at 29°C. For time-dependent induction of *brat* RNAi crosses were reared at 18°C for 6 days and then shifted at 29°C for 24 or 48 hr. For measuring cherub intensities upon *aPKC*ΔN and *miranda* RNAi expression, crosses were left for 8 days at 18°C, then shifted to 29°C for 43 hr to be able to identify separate lineages and nearest daughter cells. For survival measurements, flies were raised, collected 0–3 days after eclosion and kept at 29°C. Surviving flies were counted and moved to fresh non-yeasted vials every 2–3 days. UAS-mCherry RNAi was used as control for experiments involving UAS-transgene expression, whereas *w1118* was used as control for comparison with mutants. For temperature-induced RNAi knockdown, the driver line was crossed to *tubulin*-GAL80ts as control.

## Generation of *cherub* mutants

To create a full deletion (*cherub* DEL), FRT sites were inserted upstream and downstream of the locus using the CRISPR-Cas9 system (**Gokcezade et al., 2014**) and single-stranded oligonucleotide donors. The gRNAs used were TGGCGTCGGTTCGACCGATC and ATGAAAGTGTGAATCTTCCA.

Single-stranded oligonucleotide donors were ATCCTGGCAGACAATGGACAAAGCTCTAGCATCC TGATTGCGATCGGATCGCTTGGCGTCGGTTCGAAGTTCCTATTCTCTAGAAAGTATAGGAACTTC TGGCGGGTATATAAACTGCGGCTGCTGCGCAGAATCAATCAGTTTCATTTCAATCTTCAAACGC TGA and CTTTTACTTAACTGTGCTATTATTAAGTGAGGATATTTGGAAAAGGGATTCCAAA TGAAAGTGTGAAGTTCCTATTCTCTAGAAAGTATAGGAACTTCCAAGGGATATTTACGAAATCTG TAATAATGGTCACCACTTCTTCAAATGGTAAGAAAAAATTAA. The FRT-flanked locus was deleted by Flp-FRT recombination (*Golic and Golic, 1996*). Two gRNAs (GGCGTCGGTTCGACCGATC, GCC TGGACATGGCGCTGCG) were used to create a 180 bp deletion covering the promotor of *cherub* (*cherub* [promDEL]). Mutants were verified by Sanger sequencing and PCR.

## Generation of *cherub* [RNAi]

Generation of a *short hairpin RNA* line was performed according to the Transgenic RNAi Project's protocol (*Ni et al., 2011*). Briefly, oligonucleotides were annealed and cloned into the Walium20 vector. The oligonucleotides used were CTAGCAGTAGACATATGGTTACTGCTCGATAGTTATA TTCAAGCATATCGAGCAGTAACCATATGTCTGCG and AATTCGCAGACATATGGTTACTGCTCGA TATGCTTGAATATAAC TATCGAGCAGTAACCATATGTCTACTG.

## Oviposition assay

To assess female fecundity, five-day-old female virgins were crossed to two-day-old males in cages to allow mating for two days. The apple juice plate was replaced and flies laid eggs for 6 hr (9-15:00 each day). Subsequently, eggs were counted. The procedure was repeated on day 3 and 4 post mating. For each genotype three independent crosses were tested.

## Viability - Eclosion rate

Flies were allowed to lay eggs for 4 hr on apple juice plates. For each replicate 100 eggs were collected, transferred into fresh food vials and nine days later eclosed adult flies were counted daily for five days.

## Negative geotaxis assay

The climbing pass rate, which is the percentage of flies passing the 8.5 cm mark in 10 s, was assessed as described before (*Ali et al., 2011*). Each replicate was measured 10 times with a 1 min rest period between measurements. For each genotype and gender, 10 biological replicates consisting each of 10 two-days old adult flies were measured.

## Immunohistochemistry and antibodies

Larval brains were dissected in PBS, fixed for 20 min at room temperature in 5% paraformaldehyde in PBS and washed three times with 0.1% TritonX in PBS (PBST). After 1 hr incubation in blocking solutions (1% normal goat serum in PBST), brains were incubated with primary antibodies in blocking solution overnight, then washed three times with PBST, incubated for 2 hr at room temperature with secondary antibodies (1:500, goat Alexa Fluor, Invitrogen), washed with PBST and mounted in Vectashield Antifade Mounting Medium (Vector Labs).

Antibodies used in this study were: guinea pig anti-Deadpan (1:1000, [*Eroglu et al., 2014*]), rat anti-Asense (1:500, [*Eroglu et al., 2014*]), guinea pig anti-Syncrip (1:500, [*McDermott et al., 2012*]), rat anti-Staufen (1:100, [*Krauss et al., 2009*]), mouse anti-PH3(Ser10) (1:1000, Cell Signalling Technologies), rat anti-Elav (1:200, DSHB 7E8A10), guinea pig anti-Miranda (1:500, [*Eroglu et al., 2014*]), rabbit anti-aPKC (1:500, Santa Cruz Biotechnology), chicken anti-GFP (1:500, Abcam), rabbit anti-Imp (1:500, [*Geng and Macdonald, 2006*]).

## Fluorescence in situ hybridization (FISH)

The protocol was adapted as previously described (*Raj et al., 2008*) with minor changes. Briefly, brains were fixed for 40 min. After removing the fixative with four washes of PBS, brains were permeabilized in 70% Ethanol overnight at 4°C. Ethanol was removed and 400 µl washing buffer (2x SSC, 10% Formamide) were added. After 5 min at 37°C the wash buffer was substituted by 100 µl hybridization buffer (1 mg/ml E.coli t-RNA, 2 mM Vanadyl ribonucleoside complex, 200 nM BSA, 2x SSC, 10% Formamide, 100 mg/ml Dextran sulfate) including the FISH probes. After overnight

incubation at 37°C in the dark, brains were washed twice with washing buffer, each time for 30 min at 30°C. Brains were mounted in 2xSSC. The FISH probe sets against cherub were compromised of oligonucleotides labeled with a Quasar Dye (Stellaris Biosearch Technologies) and were designed using the Stellaris probe designer (https://www.biosearchtech.com/stellarisdesigner). DNA staining was achieved by a 20 min incubation with Hoechst (1:1000, Thermo Scientific Pierce Hoechst 33342) in washing buffer at 30°C. When antibody staining and FISH were combined, the standard immuno-histochemistry protocol was performed first, followed by the FISH protocol without incubation in 70% Ethanol. FISH probe sets against cherub were compromised of the following oligonucleotide sequences:

Against all isoforms (labeled with Quasar 570 Dye):

| | | |
|---|---|---|
| GTGTGGAGATGCTGCAACAG | GGTTGATTTTGTTCTTCTGT | CGACTGAATTTTGTTGGGCT |
| TAATCTGACATTCGGCGGTA | TGGTACTTGGGTTTCTTTTT | GCAAATTGTTCGTTTGGCTT |
| TGCCTTGTTGTGTTTAGTAT | TTTTTGCTTAAACTTAGGCT | TCGCTGACATTGATTTTTTT |
| ATGCTTTGATTTCATGTGTT | CAAGATGGTATCGTGGTTGT | GGTACTTTGGAATTTGGTTT |
| TTGGATATCAGCTTTCCTTA | TTGGTTTCAGGCGAATTACA | GCCAGATGTGTTGTTAACTT |
| ATTTGTCATCTCGTGCTATT | TGGGGTTCCTGTAAGATATA | TAGCTTTCCTTTCTTCAGAT |
| ATATGGTTACTGCTCGATCC | GAGTTTGATCCGGATGCAAG | ACAATTTCGTTGATGGGGAT |
| GCAGTAGAGAATTATGAGCA | GAATTTCGTTCTTGGCGAGA | CGATTTTGATAGCAGTTTGG |
| TTTGGCTGGTTGGGAATTTC | GCAGTTGCTTAAGACACTGG | ATTTTTTGCTTGGGGCACAG |
| GATTTCGGTGTGATTTGCTG | AATTTTGGTTTCCTTCCTTG | TATTTGGTTTGCGCCTGGAC |
| CCCCAGCAAAAGTGTACTTG | GAGTTCATCATCATCAGCTT | TCAAATGCTCGATCCTTCTC |
| CTCCTTTCTCTTGATTATCT | CTGCTTAGATATTCAGCTGG | TTTTGCTGCTTGGGGTAAAT |
| ATGTGGGCTGTATGTGTGTG | GTGTGTGCTTGTTTGTGAAT | TCGGAGAGTAAAAGGTGGGC |

Against isoform RA and RC (labeled with Quasar 570 Dye):

| | | |
|---|---|---|
| GGGGTTACTCAAATCTGTTT | TTTTACTTAGTAAGGCTCCA | GCTCGGTTTTAGGTTTTTGA |
| AGTGAAATGAGGCTCACCTT | GAAATTTTGCTTCTGGGCTG | TGGATTTTGAGTTGTGTCGG |
| GCCGTAGAAGAAGAGTTCGT | TGGAAATGAAGCCTTCTGCG | TTCAGGGATACTTTCTTTGC |
| GGCGAACTCTTTGGTTGAAT | TATTGCATTGTGTGTGTGGA | ATTCGATTTGCGTTTTGGGA |
| TGACGAAACGCGAATTTGGA | CATTTGCTTGATTTTTGCAT | GCCATGCGAAACTGTCTTTT |
| TGGTTGATTTGCGCCTAAAG | TGAGTGTATGGGTGAGCATA | AAGGGGTGTGAGTGTTTGAT |
| CTTTGCAGCCATTTTATTGC | TGGGAATCGTAGAAGCAGCT | GTCAGGCTATTTCTTTTCTT |
| TTCCTATCGAACTTTTTGGG | CCTATAACTCGCTTTCGTTG | TAGTCCTTTCGCTAACCTAT |
| TAACTGGAGTTTGGTTTGGC | AGCGCTGAGAAATGCTTGAG | GCCTTTGCTTTCTTTAATGA |
| GTGTGATTAAATAATGCCGC | ATTTCTATCAGCACTTCTGG | TGTACTGTCGTTTGCTGTAC |
| CTTAAATGACTTCCTGGGGT | TGAGTACTTAAGGGTTGGGG | GTCAATGACCAGAAGACATT |
| TTGGTCGGTCGAAGTTGTTT | TCGTAATGAGGTTTGCGTTT | GCTTGGCAGTATATTGATTT |
| ACCGGAATGTAGTATTTAGA | AGCGTGGGTAAATAAACCAA | TGTGATATCGCGTTTTGGTA |
| CGATTTTTGCAATTTCGTGA | TTTGTTTGGCTTGATTTGGA | CTTCAGTGTGTTGTCGGAAA |
| ATTTTCATGATGCATAGCGA | TTTTGAATTTGGATGCGTGA | GTGTGTGCATTATATTTTCC |
| ACCTTTTCGGATTTTGAATT | GTCAATTTTATGCTCTTTGA | TGTGACTTTTAACGGTTTGA |

Against isoform RB and RC (labeled with Quasar 570 Dye):

| | | |
|---|---|---|
| AGTTTTGATTACCTGGTCTG | CTGCTTTGTGTGATTTTGGA | CATTGGTTTGGTTGGACTTT |

*Continued on next page*

| | | |
|---|---|---|
| ACCGAAGGTTCCTCGAAATG | CTCAATTTAATGGGTGGGCT | TTTTGAATTTTTGGCGGCTC |
| CACTTCTTTTTTTCGCTCTT | TTCCGCAGTTTCTTGTAATT | TTTTTGCTCCTGACATTTGT |
| CAGAAAGTTGCCAGGAGTTT | CCGCCATTTCTAAGATTTTG | TTTCGCTGTGATTTATCTGC |
| CACGAGAGCTGGGAATTGTT | TCAGTTTCGGTTTCAGTTTT | GCAACCCTCTTCTTTGAAAT |
| GGATCAAAGTGGTCATGGTC | GGCGAAAGTAATGGGCTTGA | ATTTGTGATTTTTGTGCGGC |
| TGGGTTAAGTTGCAGTTTGT | TGCCTTCATTTTGAGTTTGA | TTAAAGTTGCGGTTCTGCTG |
| TGACACTCGTGCCGAATATT | TGCCTGGAGTTTTAAGTTTC | CCAAGTCTGACCACAATTTC |
| ATGGTTCGAACTTTTGCCAA | TTGTTTGATGTGCTTTCAGC | GCTATTGCCAGTAGTTTATA |
| ATGCCAGCCATATTTTATTT | GGCAACTTTTGCCATGAATT | ACGGTTTTTGCAGTTCTTG |
| GGTGATTTCGAGGGATTTTC | TTGGTTTTTTCTTTTTGGGC | GGTTGCTTGAATTTGCTTGA |
| CTAGCCAAAGGTGGGAAGAT | GCCCGAAACTTTTTTGGTAA | AGATTCCATTTGCTACTCTT |
| AGCTTTCTTTGCTTTTCTTT | TATACTCGCATACTACCTGT | ACCTTGTGTCCGAAATGTTA |
| TTGGGTCTAAAGAGGCCTAA | TTCCTGTTCTTTCTTCATTT | GTGGTACAATTCGCGCTTAA |
| GTTTTCCTTTCTTTAGCATT | ATTCCGCAATTCAATTGCGA | AAATGCGGCAATCGTTTCGT |
| CGCTTATCACATGTGTCGAA | TGGTTCTTTTTTTTGTCCTG | TGTTTTTGAGGTTGTACGGT |

Against isoform RC (labeled with Quasar 670 Dye):

| | | |
|---|---|---|
| GAATTTCTATCGTGCAATCA | ACCTTTTCGGATTTTGAATT | GTCAATTTTATGCTCTTTGA |
| TGTGACTTTTAACGGTTTGA | GCAGTTTTTAATTTGGTAGT | ACGCGATAATTCATTTTGCT |
| GCTGATATGTTTTTGTTTGC | ACATTGACATTTCTTTGCTT | GTATTTCTTGGTTTGAGTTT |
| ACTGCGGTTTTTGATTGATT | GTTGTTTTTGCTCTTTTTGA | GTTCGTATAATGTCTTGCTC |
| ATTCGGTTCGAACTTTTGCT | ATCTGACATTTTACTTGGTG | TTCCAATAGGGCTTTTTGTT |
| ATGTGTGTTTTGATTGTTCT | GCTATCTTCATTTTGTTTCT | GTTCATAGTTTTCAGTCTGA |
| GCCTTTTGAATTGCAATCAT | TTTTCTCAGAACTGGTTTCA | TCCTGATGCATTAAGACTCT |
| ACGTTTCCATAATTGCATAA | GTTCCGTTCTTTACCGGATA | TTTGATTTTTGTATTGCGGT |
| AGACTGGAAGTAGTTACCTT | GTAAAATGGTGGCACATGTT | TTTGGTTTTCTAGACCATTT |
| TTTTGAGATTCAAATCGGGT | GTGTTATTTAAGGCATTTGG | ATTTGGTTTGTCTGGTTCTA |
| CAGAGGAGCCGAAAGTTGAA | GGTTTTCGGTTTTCAAATTT | |

For confirming RNA specificity of the probe set against cherub RNA, the above mentioned standard protocol was performed except that brains were incubated for 2 hr at 37°C with RNase A (100 µg/ml) before the incubation step in 70% Ethanol or with RNase H (100 U/ml) after hybridization in order to let RNA/DNA duplexes be formed first.

## In vitro FISH/Immunofluorescence

Cells from 25 to 35 dissociated larval brains (UAS-*dcr2*; *wor*-GAL4, *ase*-GAL80; UAS-CD8::GFP) were plated on cell culture dishes and incubated in complete Schneider's medium (*Homem et al., 2013*) with Colcemid (25 µM) for 5 hr at 25°C. After a 15 min fixation with 5% PFA in PBS at room temperature, cells were incubated with 3% normal goat serum in 0.1% TritonX in PBS overnight, one hour with primary antibodies and one hour with secondary antibodies. Between steps cells were washed with PBS. Cells were again fixed for 20 min, washed with FISH washing buffer and FISH was performed with a probe incubation time of 4 hr at 37°C. After one wash with washing buffer for 30 min at 37°C, cells were imaged in 2xSSC. NBIIs were unambiguously identified by a size >10 µm and GFP expression.

## Microscopy and image analysis

Confocal images were acquired with Zeiss LSM 780 confocal microscopes. For intensity measurements of cherub foci, dot areas were marked on the plane of a z-stack with the largest diameter. Raw integrated density (sum of grey values of all selected pixels) was measured using FIJI. To

determine intensity ratios in *Figure 6G and I*, cytoplasmic cherub of interphase NBIIs and closest daughter cells were measured as raw integrated intensities (I) using FIJI. Ratios were calculated as (I $_{daughter\ cell}$/area $_{daughter\ cell}$) / (I $_{NBII}$/area $_{NBII}$). For intensity quantifications in cross sections of cells, FIJI's plot profile function was used and the cross section length of each cell was scaled to 100%. Cortical cherub intensity in cells of 72 hr clones were measured by determining the cortex via cortical Miranda staining and measuring raw integrated intensities, which were normalized to the cortical area.

For the 3D movie, Imaris was used to 3D reconstruct a z-stack of one NBII from an in vitro FISH/IF experiment. To measure the relative position of cherub/Syp and aPKC crescents, a perpendicular line was drawn through the center of each crescent, which was defined by bisecting the connecting line between the crescent's ends (see scheme in *Figure 6E*). The angle between the two bisecting lines was measured using FIJI. An angle of 180°C is measured when cherub/Syp and aPKC form crescents opposite of each other (this is equivalent to a cherub/Syp crescent perpendicular to the mitotic spindle). For quantifications of tumor volumes, one brain lobe per brain was imaged and used for quantifications.

To evaluate colocalization, z-stacks were recorded of in vitro mitotic NBIIs using 63x immersion oil objective with optimal pixel size and z-stack distance. Pearson's coefficient and Li's intensity correlation analysis (ICA) were calculated using the FIJI plugin JACoP (*Bolte and Cordelières, 2006*). The covariance of intensity is calculated as (pixel intensity Ai – mean intensity A)(pixel intensity Bi – mean intensity B), A and B being the respective channels. ICA plots were made in R.

## Statistics

Statistical analyses were performed with GraphPad Prism 7. Unpaired two-tailed Student's *t*-test was used to assess statistical significance between two genotypes/conditions and one-way ANOVA for comparison of multiple samples. No statistical methods were used to predetermine the sample size. Sample sizes for experiments were estimated based on previous experience with a similar setup that showed significance. Experiments were not randomized and investigator was not blinded.

## RNA-Immunoprecipitation (RIP)

RIP was performed as previously described (*Gilbert and Svejstrup, 2006*) with minor modifications. Shortly, dissected brains were cross-linked with 0.5% Formaldehyde, quenched with Glycin (final 0.125 M) and homogenized in RIP lysis buffer (50 mM Hepes pH 7.5, 140 mM NaCl, 1 mM EDTA, 1% Triton X-100, 0.1% Sodium-deoxycholate, 1x protease inhibitor, 1 mM PMSF, 40 U/ml RNasin). After DNA removal using DNaseI, protein lysates were incubated overnight with antibodies (goat anti-Staufen dN-16 antibody, Santa Cruz Biotechnology or mouse anti-HA antibody clone 12CA5, in both cases 10x blocking peptide was used) at 4°C and subsequently incubated with Protein G Dynabeads (Thermo Scientific Fisher) for 1 hr. Eluted Protein-RNA complexes were treated with Proteinase K. After RNA extraction, pull-down of RNAs was detected by qPCR.

## RNA isolation, cDNA synthesis and qPCR

RNA was isolated by using TRIzol reagent (Ambion) or acidic Phenol/Chloroform (5:1, pH4.5, Ambion) for RIP samples. Ethanol-precipitated RNA samples were then used as template for first-strand cDNA synthesis with random hexamer primers (SuperScriptIII, Invitrogen). qPCR samples were prepared with Bio-Rad IQ SYBR Green Supermix and run on a Bio-Rad CFX96 cycler. Primers used were:

 *cherub* - all isoforms (AGCAGCACCAGCAGCAGTAG, GCGGTGGATTTGGTTGATTT),
 *cherub* - RC isoform (TCAAAAGGCGATGAAACCAGT, ATTGCGGTTTGTTCCGTTCT),
 *brat* (CACAAGTTCGGGTGCTCTAA, CCGATTGTCGCTGATGAAGA),
 *sle* (GAGTCCGTTGGCAGTAAAGATA, CTCGTCTTCGTTGTCCGATAC),
 *CG42232* (GAAGATGGCGGTGAAGTAGAA, GGCCTGTAGAGCTGGAATTAG),
 *CycG* (CACTACACTCACCCTTGATTCC, CGAGTTGTACGAAACCCTCAA),
 *Gbs-76A* (GTCCACATCTACGGTGAGATTAC, ACCAGAGGAAAGCAGGAATG),
 *CG13185* (GTCTGGAGTTCGATCAGGAAAG, GTCGGAAGCATCTGGTGTATAG),
 *spen* (GAAGAGCGGCATCGACTAAA, GGCAAAGAAGGTGAGGTAGAA),
 *Su(var)2-HP2* (TCCTTGGGATTCGGGAAATG, GAGGCTGCTACTGAGCTAATG),

*tai* (ACTACGGTGGCTTCAACTTC, TGGATTGCTACTGCTGCTATT),
*CG32479* (GCAAGTCCCACAGCAACTAT, CTGCGGATTGGCTGATGAA),
*Su(Tpl)* (GGTACTCATCGTAGTCGCTTTC, CGCTACGACTTCAGCCAATA),
*Taf12L* (ACAGCGATAAATCGTCGGATAA, GGACAGACTGGCTCTCAATTAC),
*CG41128* (TTAAAGGATGTGGAGGCGTAAT, TCCTATAAGCGATGCCCATTC),
*Hsp67Ba* (GCCAGCAATCTCCCACTATT, AATAATCTGCACGGGTAGGC),
*CG2021* (CATGAGCGCGTCTTCTCTAC, AGTCGATGGTCTCGTCTATCA),
*CG11882* (ACTAACAGCGTCAGCTTCTC, GAGCCTGATGAAGGGCTATT).
Gene expression was normalized to *Act5C*
(AGTGGTGGAAGTTTGGAGTG, GATAATGATGATGGTGTGCAGG).
Primers used for RIP-qPCR were:
*RpL32* (GCCGCTTCAAGGGACAGTAT, TTCTGCATGAGCAGGACCTC),
*cherub* - all isoforms (AGCAGCACCAGCAGCAGTAG, GCGGTGGATTTGGTTGATTT).
Primers for RT-PCR (30 cycles) used to detect expression in different *Drosophila* species were:
for *GPDH*:
*D. melanogaster* (AACTTCTGCGAAACGACAAT, CGTAACACGTCGTGATCAG),
*D. simulans* (AACTTCTGCGAAACGACAAT, CGTAACACGTCGTGATCAG),
*D. willistoni* (ATACCATGCGCCGTACTG, CATAACACGTCGTGATAAGATCC),
for *cherub*
*D. melanogaster* (AGCAGCACCAGCAGCAGTAG, GCGGTGGATTTGGTTGATTT),
*D. simulans* (GAGTAGGAGCCGCACAGGAG, CGGTGTGGAGATGCTGCAAC),
*D. willistoni* (GGAAGGATCTATGCAGAGAGACA, CCCCAACCTTCTTGTGTCCG).

## Immunoprecipitation and western blotting

Immunoprecipitation was performed according to the RIP protocol except omission of formaldehyde fixation, DNaseI treatment steps and instead of RNA isolation samples were boiled in 2x Laemmli buffer to elute protein complexes and loaded on 3–8% gradient Tris-Acetate gels (NuPAGE, Invitrogen). After SDS-PAGE according to Invitrogen's protocol, proteins were transferred to a Nitrocellulose membrane (0.22 µm, Odyssey LI-COR) for 2 hr at 100V, blocked with 5% milk powder in blocking solution (PBS with 0.2% Tween) for 1 hr, overnight incubation with primary antibody in blocking solution at 4°C, 3x washed with washing solution (PBS with 0.1% Tween) and followed by 1 hr incubation with secondary antibody (HRP-linked Whole Antibodies from GE Healthcare) in blocking solution. After three washes with washing solution, horseradish peroxidase activity was detected with Pierce ECL Plus (Thermo Fisher Scientific). Antibodies used were: goat anti-Staufen (1:2000, dN-16 Santa Cruz Biotechnology), guinea pig anti-Syncrip (1:3000, (*McDermott et al., 2012*)), mouse anti-HA (1:500, clone 12CA5).

## PhyloCSF analysis

A Multiz alignment of 27 insecta (23 *Drosophila* species, house fly, *Anopheles gambiae* and *mellifera*, honey bee and *Tribolium castaneum*) aligned to *Drosophila* genome dm6 in multiple alignment format (MAF) was downloaded from UCSC Genome Browser. The strand specific gene models of the mentioned genes (FlyBase r6.09) were provided as BED files. Those inputs were used in Galaxy (*Blankenberg et al., 2010*) in 'Stitch MAF blocks' followed by 'concatenate FASTA alignment by species' functions to generate FASTA alignments for each gene in the 12 *Drosophila* specified by the PhyloCSF phylogeny. PhyloCSF (*Lin et al., 2011*) was run with the resulting FASTA file using the following parameters: '–orf=ATGStop –frames=3 removeRefGaps –aa'.

## Transplantations

Crosses were set up at 29°C. Third-instar larval brains were collected after 5–6 days, NBs were isolated by FACS (*Harzer et al., 2013*) and transplantations of GFP[+] NBs (UAS-*dcr2; insc*-GAL4, UAS-*stinger*::GFP) or RFP[+] tNB (*brat*[RNAi] driven by UAS-*dcr2; wor*-GAL4, *ase*-GAL80; UAS-*stinger*::RFP) suspensions were performed as previously described (*Caussinus and Gonzalez, 2005*) with minor modifications. Pictures of transplanted host flies were taken with a Sony Alpha NEX-5 compact camera.

## DNA content analysis

Dissected brains were enzymatically dissociated in Rinaldini solution as described previously (*Berger et al., 2012*) and incubated with Hoechst 33342 (20 µM) for 1 hr at room temperature. Then samples were kept on ice until FACS sorting. Data plots were generated with FlowJo software.

## Sample preparation for whole-genome sequencing

Genomic DNA was isolated from the dissected tumor brains and abdomens of the same adult female *brat* [k06028] fly using standard Phenol-Chloroform extraction procedure including RNase and Proteinase K treatment. In total, three flies were sequenced. DNA was fragmented using a microtip sonicator (Omni-Ruptor 250, Omni International). Quality control was performed with Agilent High Sensitivity DNA Kit (Agilent Technologies). DNA libraries were prepared using NEBNext Ultra DNA Library Prep Kit (Illumina) and 100 base pair paired-end sequencing was performed on a Hiseq2000 platform. After deduplication, an average sequencing depth of >170 x was achieved for each sample. Sequence data has been deposited at the short read archive (https://www.ncbi.nlm.nih.gov/sra, SUB1954694).

## Analysis of whole-genome sequencing data

Leading and trailing Ns of the paired reads were trimmed. Reads were aligned with BWA (v0.6.2) (*Li and Durbin, 2009*) to the genome (FlyBase r5) with a maximum insert size of 1000. Picard tools (v1.82, http://broadinstitute.github.io/picard) were used to fix the alignment (CleanSam) and add read groups (AddOrReplaceReadGroups). Duplicates (MarkDuplicates) were marked within all samples derived from the same fly. Reads of a fly were realigned with GATK (v2.3) for each chromosome and remerged with Picard tools. Summary statistics were computed with GATK (*McKenna et al., 2010*). Somatic point mutations were identified with MuTect (v1.1.4) (*Cibulskis et al., 2013*). InDels (strand.bias = TRUE) were identified with the SomaticIndelDetector of GATK. Variants were characterized with SnpEff (v3.2a) (*Cingolani et al., 2012*). For coverage plots, reads were counted in genomic bins and normalized by the median. The foldchange and coverage were plotted with R. Published aCGH datasets were used to identify under-replicated (*Sher et al., 2012*) and amplified (*Kim et al., 2011*) regions.

## RNA sequencing – DigiTAG

We devised a method that combines transposon-mediated library preparation with molecular barcoding to quantify the original library molecules rather than their amplicons (*Figure 3B*). The NBII driver line UAS-*dcr2; wor*-GAL4, *ase*-GAL80; UAS-*stinger*::RFP was used. Brains from wandering third instar larvae were dissected, dissociated and NBs were isolated by FACS (*Berger et al., 2012*; *Harzer et al., 2013*). For each condition, three samples were prepared. RNA from 300 cells per sample was used for library preparation. Total RNA isolated with TRIzol LS reagent (Ambion) was reverse transcribed into first strand cDNA using Superscript III Reverse Transcriptase (Invitrogen) with oligo-(dT)20 primers. After second strand synthesis was performed, the sequencing library was prepared with the Nextera DNA Library Preparation Kit (Illumina). In an enzymatic tagmentation reaction, cDNA was simultaneously fragmented and tagged with adapter sequences: 15 µl TDE1 Tagment DNA buffer, 0.2 µl TDE1 Tagment DNA enzyme was added to 15 µl cDNA and incubated for 5 min at 55°C. After purification (Agencourt AMPure XP beads, Beckman Coulter), 19.5 µl tagmented DNA was PCR amplified using 25 µl Phusion HF 2x master mix (Thermo Fisher Scientific), 2.5 µl 20x Eva Green (Biotium), 1 µl Nextera primers mix (10 µM each), 1 µl Index two primers (N501-N506, for multiplexing) and 1 µl modified Index one primers, which included random 8-mer tags for molecular barcoding. Cycling conditions according to the manufacturer (Nextera DNA Library Preparation Kit, Illumina) were used. Purified libraries (Agencourt AMPure XP beads) were subjected to 50 base pair Illumina single-end sequencing on a Hiseq2000 platform.

## Transcriptome data analysis

The reads were screened for ribosomal RNA by aligning with BWA (v0.6.1) (*Li and Durbin, 2009*) against known rRNA sequences (RefSeq). The rRNA subtracted reads were aligned with TopHat (v1.4.1) (*Trapnell et al., 2009*) against the *Drosophila melanogaster* genome (FlyBase r5.44) and a maximum of 6 mismatches. Introns between 20–150000 bp were allowed which is based on FlyBase

statistics. Maximum multihits was set to one and InDels as well as Microexon-search was enabled. Additionally, a gene model was provided as GTF (FlyBase r5.44). snRNA, rRNA, tRNA, snoRNA and pseudogenes were masked for downstream analysis. Reads arising from duplication events were marked as such in the alignment (SAM/BAM files) as follows: The different tags were counted at each genomic position. Thereafter, the diversity of tags at each position was examined. First, tags were sorted descending by their count. If several tags had the same occurrence, they were further sorted alphanumerically. Reads sharing the same tag, were sorted by the average PHRED quality. Again if several reads had the same quality, they were further sorted alphanumerically. Now the tags were cycled through by their counts. Within one tag, the read with the highest average PHRED quality was the unique correct read and all subsequent reads with the same tag were marked as duplicates. Furthermore, all reads which had tags with one mismatch difference compared the pool of valid read tags were also marked as duplicates. The aligned and deduplicated reads were counted with HTSeq (*Anders et al., 2015*; *Li et al., 2009*) and the polyA containing transcripts were subjected to differential expression analysis with DESeq (v1.10.1) (*Anders and Huber, 2010*). Note that the basemean values of *brat* mRNA appear unchanged in *brat* [RNAi] NBII compared to control NBII. We attribute this to a large number of reads, unique to the *brat* [RNAi] condition, matching non-coding regions in the second intron of *brat-RA, -RE*, and the first intron of *brat-RB, -RC*. Conversely, reads matching the coding region of the gene are reproducibly lower in *brat* [RNAi] condition. Furthermore, *brat* knockdown was confirmed by qPCR targeting a coding region. Data has been deposited in the data depository Gene Expression Omnibus (http://www.ncbi.nlm.nih.gov/geo/, GEO serial accession number GSE87085). To investigate the expression of early temporal NB identity genes (*Figure 8A*) in tNBs we made use of a previously published gene dataset from antenna lobe NBs (*Liu et al., 2015*).

## Sequence identity, conservation and RNA structure analysis

The Multiz alignment of 27 insects was accessed for the genomic locus via the UCSC table browser. The MAF blocks are stiched in Galaxy (*Blankenberg et al., 2011*). Percent of sequence identity defined as '100 * (identical positions) / (aligned positions + internal gap positions)' was calculated with Biostrings in R (R package version 2.40.2.). *Drosophila suzukii* was excluded due to a very small homology region. Thermodynamically stable and evolutionary conserved RNA structures were predicted using the RNAz Web server (*Gruber et al., 2007*) using step size 10 and a window size of 200. Sequence repeats were identified by RepeatMasker (*Jurka, 2000*) and conservation was assessed by PhastCons and PhyloP (*Siepel et al., 2005*) via the UCSC Genome Browser (dm6).

## Acknowledgement

We thank all Knoblich lab members for support and discussions, Julius Brennecke, Francois Bonnay, Christopher Esk and Josh Bagley for comments on the manuscript, Peter Duchek, Joseph Gokcezade, Elke Kleiner, the IMP/IMBA Biooptics Facility and the Next Generation Sequencing Unit of the Vienna Biocenter Core Facilities (VBCF) for assistance and A Ephrussi, I Davis, Y Hiromi, P Macdonald, T Lee, the Harvard TRiP collection, the Bloomington Drosophila stock center and the Vienna Drosophila Resource Center (VDRC) for reagents.

## Additional information

### Funding

| Funder | Grant reference number | Author |
| --- | --- | --- |
| Austrian Academy of Sciences | | Jürgen Knoblich |
| European Commission | | Jürgen Knoblich |
| Austrian Science Fund | Z_153_B09 | Jürgen Knoblich |
| European Molecular Biology Organization | | Francois Bonnay |

The funders had no role in study design, data collection and interpretation, or the decision to submit the work for publication.

### Author contributions
Lisa Landskron, Conceptualization, Formal analysis, Investigation, Visualization, Writing—original draft, Project administration; Victoria Steinmann, Heike Harzer, Anne-Sophie Laurenson, Investigation; Francois Bonnay, Investigation, Visualization; Thomas R Burkard, Formal analysis; Jonas Steinmann, Methodology; Ilka Reichardt, Resources; Heinrich Reichert, Supervision; Jürgen A Knoblich, Conceptualization, Supervision, Funding acquisition, Writing—original draft, Project administration

### Author ORCIDs
Lisa Landskron (iD) http://orcid.org/0000-0003-0405-2046
Jürgen A Knoblich (iD) https://orcid.org/0000-0002-6751-3404

### Decision letter and Author response
Decision letter https://doi.org/10.7554/eLife.31347.033
Author response https://doi.org/10.7554/eLife.31347.034

# Additional files

### Supplementary files
• Transparent reporting form
DOI: https://doi.org/10.7554/eLife.31347.022

### Major datasets
The following datasets were generated:

| Author(s) | Year | Dataset title | Dataset URL | Database, license, and accessibility information |
|---|---|---|---|---|
| Lisa Landskron, Jürgen A Knoblich, Thomas R Burkard | 2017 | brat tumor sequencing | https://www.ncbi.nlm.nih.gov/sra/?term=SRP090130 | Publicly available at the NCBI Sequence Read Archive (accession no. SRP090130) |
| Francois Bonnay, Thomas R Burkard, Jürgen A Knoblich | 2017 | NBII vs tNB transcriptome | https://www.ncbi.nlm.nih.gov/geo/query/acc.cgi?acc=GSE87085 | Publicly available at the NCBI Gene Expression Omnibus (accession no. GSE87085) |

The following previously published datasets were used:

| Author(s) | Year | Dataset title | Dataset URL | Database, license, and accessibility information |
|---|---|---|---|---|
| Sher N, Bell GW, Li S, Nordman J, Eng T, Eaton ML, Macalpine DM, Orr-Weaver TL | 2012 | CGH to ascertain levels of gDNA in third instar salivary glands of various mutant Drosophila | https://www.ncbi.nlm.nih.gov/geo/query/acc.cgi?acc=GSE31900 | Publicly available at the NCBI Gene Expression Omnibus (accession no. GSE31900) |
| Kim JC, Nordman J, Xie F, Kashevsky H, Eng T, Li S, MacAlpine DM, Orr-Weaver TL | 2011 | Input DNA from OregonRTOW Stage 10 egg chambers | https://www.ncbi.nlm.nih.gov/geo/query/acc.cgi?acc=GSE29517 | Publicly available at the NCBI Gene Expression Omnibus (accession no. GSE29517) |

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
