## [Decision Letter]

Thank you for submitting your article "The asymmetrically segregating lncRNA cherub is required for transforming stem cells into malignant cells" for consideration by *eLife*. Your article has been favorably evaluated by K VijayRaghavan (Senior Editor) and three reviewers, one of whom, Hugo Bellen, is a member of our Board of Reviewing Editors. The reviewers have opted to remain anonymous.

The reviewers have discussed the reviews with one another and the Reviewing Editor has drafted this decision to help you prepare a revised submission.

The paper addresses an interesting question in tumor biology: How do tumor cells become immortal? The authors show that a lncRNA (cherub) is upregulated in tumorous neuroblasts compared to wild type, and that it plays a crucial role in the immortality of tumor cells. Interestingly, this lncRNA is dispensable for normal development and the flies are fully fertile and healthy. The authors also present evidence that the cellular localization of cherub during neuroblast divisions is the critical factor for cherub to be tumorigenic in a brat background. cherub is localized asymmetrically in dividing neuroblasts where it is enriched at the cell cortex, and it is inherited in daughter cells where it is cytoplasmic in newborn intermediate precursors of Type II neuroblasts. The gene is increased in brat mutant neuroblasts, which leads to increased expression in INP, which might return to neuroblast fate. Two RNA binding proteins Syp and IMP have been proposed to regulate the age of the neural stem cells by others, with opposing gradients over time. Here they show that IMP and Syp have some altered levels/distribution in the tumors from loss of Brat; this is less clear for Syp than for IMP. They propose that Syp is involved in linking Syp to Staufen: they detect binding of Syp to staufen in RIP, which is reduced in the absence of cherub. The resulting model is that, by linking Staufen to Syp, cherub contributes to the mechanism of neural stem cell aging, so that in its absence the NBs/tNBs adopt a younger character more prone to tumors. The model is an attractive one, but not all the supporting data are clear-cut nor is the fit between the model and the data straightforward. However, the study is significant to the field of tumor biology, as well as for a more general understanding of the molecular function of lncRNAs.

Comments:

1) The authors are careful to use two deletions and an RNAi line for cherub, but a genomic rescue construct should restore tumor formation in brat/cherub mutants. The authors should show this control.

2) Is overexpression of cherub sufficient to drive tumorigenesis or increase proliferation? Please provide data.

3) Failure to detect a peptide in proteomic analyses does not necessarily mean the peptide is not made. How confident are the authors that no small peptides exist for cherub? See Zanet J, et al. (2015) Science 349:1356-1358 for another outcome.

4) Figure 6: While one can identify cortical cherub in Lgl3A, it is difficult to detect differences in neuroblasts of the other three panels, especially in print. Similarly, in Figure 6, accumulation of cherub in staufen knockdown neuroblasts is obvious, but no evidence of asymmetric segregation failure. This should be more convincingly shown.

5) The authors mention that "Small insertions and deletions (InDels) were mainly restricted to introns or intergenic regions" and consider such changes as insignificant in terms of tumorigenesis. However, we know that changes in regulatory regions could be meaningful as they could be templates for transcription factors or ncRNAs.

6) Subsection “Tumor neuroblasts show defects in temporal neuroblast identity and require early temporal factors”, first paragraph, shouldn't it be tNBs instead of NBIIs? It seems like despite tNBs being collected at late larval stages, they still express genes usually restricted to early NBs. The authors argue that cortical localization of cherub is, "enhanced in brat tNBs" – this is not evident in Figure 7. Likewise that "extra tNBs had strongly elevated cortical cherub" is also not clear in Figure 7. Proper quantification of these results are needed as they are pivotal to their model. Please address.

7) Subsection “Tumor neuroblasts show defects in temporal neuroblast identity and require early temporal factors”, first paragraph, the authors refer to Figure 8. The way this figure is presented is not very clear. The authors write in the figure legend "see Materials and methods". However, I cannot find where in the Materials and methods part this is described, and what methods were used for the evaluation.

8) There appears to be a gap in the model, the authors do not really explain how/why the presence of cherub "rejuvenate" the NBs via its effects on Syp. This is a key point that the authors need to address. For example, cherub is proposed to bind to Syp linking it to Staufen so that it segregates correctly. Knock down of cherub perturbs Syp segregation, but gives the opposite phenotypes (Syp KD enhances the tumors, cherub KD reduces them). A clearer explanation for their interpretation of these data is needed. The model also makes the prediction that Staufen KD should have the same effects on tumorigenesis as cherub, is this the case? What happens if both Syp and cherub are knocked down at the same time?

9) The effects of removing Staufen on Syp are much more pronounced than removing cherub, how do the authors explain this difference (Figure 9).

10) Was Imp also detected in the RIP? Imp also is affected by cherub knock-down and the phenotypes of IMP KD are more similar to cherub KD. How do they explain this? More experiments or at least discussion of the relationship with IMP are needed.

11) If cherub is involved in regulating the temporal state of the stem cells, is the expectation that early induced brat tumors would not be suppressed by cherub KD? The authors should investigate effects at earlier stages to verify their model.

12) The model implies that cherub, Staufen and Syp will co-localize in the crescent, but this has not been shown.

---

## [Author Response]

Comments:1) The authors are careful to use two deletions and an RNAi line for cherub, but a genomic rescue construct should restore tumor formation in brat/cherub mutants. The authors should show this control.

We have generated transgenic flies harboring a cherub genomic rescue construct. Using this line, we were able to revert the tumor rescue phenotype of cherub. Our observation is that the rescue construct can increase the number of Dpn+ tNBs in in brat^RNAi^ cherub ^DEL-/-^ brains. These data are now included in Figure 5—figure supplement 1. Importantly, the new data not only verify the specificity of our mutant alleles but they also demonstrate that cherub can exert its tumor-suppressive function when inserted at another genetic location.

2) Is overexpression of cherub sufficient to drive tumorigenesis or increase proliferation? Please provide data.

We agree that the overexpression of cherub would be very informative. We have made transgenic flies overexpressing the longest or the shortest cherub isoform. Cherub expression from this inducible transgene using the UAS/GAL4 system did not lead to the formation of stable cherub transcripts in the cytoplasm or at the cortex, although we detected an additional nuclear dot with FISH probes against cherub indicative of successful transgene transcription (see Author response image 1). We would like to mention that this phenomenon has been also described for other temporal factors. It has been shown for both, Imp and Syp, that expressing them from an overexpression construct in NBs only weakly – if at all –increases expression (Liu et al., 2015; Yang et al., 2017a). Thus, the temporal identity programs in central brain NBs are tightly regulated on a post-transcriptional level.

**Author response image 1. respfig1:** Overexpression of cherub RA isoform with UAS-dcr2; wor-GAL4, ase-GAL80; UAS-CD8::GFP. The combination of isoform-specific FISH probes allows the detection of the transgene in the nucleus (arrow). Thus, the transgene is induced in NBs despite the lack of elevated cherub levels in the cytoplasma/cortex. Scale bars 10 μm.

3) Failure to detect a peptide in proteomic analyses does not necessarily mean the peptide is not made. How confident are the authors that no small peptides exist for cherub? See Zanet J, et al. (2015) Science 349:1356-1358 for another outcome.

Additionally, to proteomic analysis we showed that cherub isoforms have a negative (non-coding) maxCSF score, a measurement that also takes into account the evolutionary signature of 12 *Drosophila* species and has been used in other studies to assess the coding potential of lncRNAs (e.g. (Lee et al., 2016; Li et al., 2012)). Likewise, coding potential calculator, an algorithm that has been shown to correctly predict the coding potential of the polished rice RNAs mentioned by the reviewer (Ríos-Barrera et al., 2015), assigns negative hit scores (non-coding) to all cherub isoforms (RA:-0.86, RB-0.89:, RC:-0.92). Furthermore, we show a function for the cherub RNA (binding between Staufen and Syncrip) and its presence in complexes involved in the proposed function. Altogether, this strongly suggests a non-coding function, although we cannot formally rule out that also peptides are made. In the main text we already suggested “Thus, CR43283 likely acts as a lncRNA.” and now we modified the text further to “Peptides of short open reading frames found in CR43283 were not detected in the brat tumor proteome and PhyloCSF analysis of CR43283 did not reveal any coding potential” – please see subsection “The lncRNA cherub is upregulated in tumor neuroblasts compared to type II neuroblasts”.

4) Figure 6: While one can identify cortical cherub in Lgl3A, it is difficult to detect differences in neuroblasts of the other three panels, especially in print. Similarly, in Figure 6, accumulation of cherub in staufen knockdown neuroblasts is obvious, but no evidence of asymmetric segregation failure. This should be more convincingly shown.

To address this point, we reasoned that if cherub is distributed symmetrically during mitosis, the resulting two daughter cells (NBII and INP) should have similar amounts of cherub. Whereas if cherub is cortical and thus asymmetrically segregated, cherub levels should be higher in the most recently born INP when compared to the NBII. Indeed, measuring cytoplasmic cherub intensity upon loss of staufen revealed equal levels in NBII and surrounding INP, confirming that Staufen is important for asymmetrically allocating cherub. These results are presented in Figure 6. In addition, we also provide now images of a staufen ^RNAi^ NB in mitosis, which does not show a basal cherub crescent. Please see Figure 6—figure supplement 2.

Similarly, we measured cytoplasmic cherub intensities in NBIIs and daughter cells upon *miranda*^RNAi^ or aPKCΔN expression. Under both conditions, we detected equal signal intensities for cherub in the two daughter cells indicating similar cherub concentrations in both cells. These data strengthen the results in Figure 6 and show that the canonical asymmetric cell division machinery is required for cherub’s asymmetry.

5) The authors mention that "Small insertions and deletions (InDels) were mainly restricted to introns or intergenic regions" and consider such changes as insignificant in terms of tumorigenesis. However, we know that changes in regulatory regions could be meaningful as they could be templates for transcription factors or ncRNAs.

We apologize if it may have sounded as if we disregard InDels found in introns and intergenic regions due to their nature. Rather we suggest that we did not find recurrent DNA changes (including InDels) as we did not identify a region (defined as 10000bp), that harbored InDels in all tumor samples. Therefore, it is unlikely an element common to all tumors is affected by this InDels. We have modified the text to clarify this.

6) Subsection “Tumor neuroblasts show defects in temporal neuroblast identity and require early temporal factors”, first paragraph, shouldn't it be tNBs instead of NBIIs? It seems like despite tNBs being collected at late larval stages, they still express genes usually restricted to early NBs. The authors argue that cortical localization of cherub is, "enhanced in brat tNBs" – this is not evident in Figure 7. Likewise that "extra tNBs had strongly elevated cortical cherub" is also not clear in Figure 7. Proper quantification of these results are needed as they are pivotal to their model. Please address.

Yes, correctly it should be tNBs. We have corrected this mistake (subsection “Tumor neuroblasts show defects in temporal neuroblast identity and require early temporal factors”) and also included now quantifications to make the data in Figure 7 more evident. We measured cherub levels in cross sections of cells. These measurements show that in brat tNBs cortical cherub is elevated. This data can be found in Figure 7—figure supplement 1. It seems that Figure 7 is confusing as the point we wanted to make here is that the tNBs have increased cortical cherub compared to the NBII. We provide 72h instead of 48H clones since at 72h more tNBs have reached a NB-like size and this will guide the reader better to compare it to the correct wildtype cell, which is the NBII. Furthermore, we quantified cortical cherub levels in these NBs to clearer show the accumulation of cherub in tNBs – please see new Figure 7.

7) Subsection “Tumor neuroblasts show defects in temporal neuroblast identity and require early temporal factors”, first paragraph, the authors refer to Figure 8. The way this figure is presented is not very clear. The authors write in the figure legend "see Materials and methods". However, I cannot find where in the Materials and methods part this is described, and what methods were used for the evaluation.

The same data are now depicted in a new graph and we hope it is now easier to understand. We removed the “see Materials and methods” as it leads to confusion. The analyses of differential coverage and the reference for the list of early NB identity genes can be found at the very end in the Materials and methods part “Transcriptome data analysis”.

8) There appears to be a gap in the model, the authors do not really explain how/why the presence of cherub "rejuvenate" the NBs via its effects on Syp. This is a key point that the authors need to address. For example, Cherub is proposed to bind to Syp linking it to Staufen so that it segregates correctly. Knock down of cherub perturbs Syp segregation, but gives the opposite phenotypes (Syp KD enhances the tumors, cherub KD reduces them). A clearer explanation for their interpretation of these data is needed. The model also makes the prediction that Staufen KD should have the same effects on tumorigenesis as cherub, is this the case? What happens if both Syp and cherub are knocked down at the same time?

Our data suggest that the loss of cherub leads to enriched cytoplasmic Syp in tNBs and a reduction in tumor growth, while the depletion of Syp in tNBs has the opposite effect and enhances tumor growth. These results suggest that Syp promotes tumor growth. cherub might inhibit Syp directly by binding to it and/or indirectly by removing Syp from the tNB. Syp has been shown to bind many different mRNA targets and to regulate their translation either negatively or positively (McDermott et al., 2014). The direct targets of Syp in the brain are currently unclear. However, genetic experiments showed that Syp represses early NB factors like Imp (Syed et al., 2017; Yang et al., 2017a) and also promotes the expression of genes restricting NB proliferation (Yang et al., 2017a; 2017b). cherub is likely to inhibit the latter one for two reasons: 1) We have shown that in brat tumors high cherub levels recruit Syp to the cortex, but only a minor subset expresses Imp. 2) We performed immunostainings showing that cherub mutant NBIIs do not show a change in expression or timing of Imp (data not shown). In the subsection “cherub regulates Syp locatization in brat tumors”, we now provide an extensive discussion to explain this.

We have tested the knock down of staufen in brat tumors and could not see differences in tumor growth. At the current state, we can only speculate why this is. Importantly, in a cherub knockdown situation NBs have high cytoplasmic levels of Syp, whereas in a staufen knockdown condition NBs have high cytoplasmic cherub and Syp. This indicates that cherub inhibits Syp function.

Unfortunately, due to difficulties in generating the appropriate fly stocks, it was not possible to perform Syp cherub double knockdown experiments in a brat tumor background within a reasonable time frame.

9) The effects of removing Staufen on Syp are much more pronounced than removing cherub, how do the authors explain this difference (Figure 9).

Our interpretation of these results is that in both of these conditions, the neuroblast shows enriched cytoplasmic cherub and – unlike in control NB lineages – cherub is not enriched in the latest born INP. This indicates a failure to distribute Syp asymmetrically between NBII and INP. We did recognize that cherub levels are decreased in the daughter cells of staufen depleted NBs. It should be noted that staufen has been shown to bind many RNAs and thus might show a more complex phenotype. Staufen depleted NBs appear less healthy in general: They show a deformed NB shape and have irregular nuclei with membrane protrusions into the nucleus, which can be best seen in Figure 6. A delay in NB division would explain the different cherub concentrations in NB and INP as while cherub is readily degraded in INPs it accumulates in NBs.

10) Was Imp also detected in the RIP? Imp also is affected by cherub knock-down and the phenotypes of IMP KD are more similar to cherub KD. How do they explain this? More experiments or at least discussion of the relationship with IMP are needed.

We have preliminary data suggesting that Imp might co-immunoprecipitate with Staufen as well. However, NBs of cherub mutants did not show any defects in Imp localization or in the timing of Imp expression. In contrast, Syp levels within the NB lineage clearly depend on the Staufen-cherub complex and were altered in cherub mutants. For those reasons, we did not investigate Imp further. The relationship between Imp, Syp and cherub is now discussed in more detail in the Discussion of the manuscript. See subsection “cherub regulates Syp locatization in brat tumors” and comment #8.

11) If cherub is involved in regulating the temporal state of the stem cells, is the expectation that early induced brat tumors would not be suppressed by cherub KD? The authors should investigate effects at earlier stages to verify their model.

We think this comment is due to a misunderstanding as all our tumors are induced early and we would like to clarify this:

The Maurange lab has shown that an early tumor induction is required to develop malignant tumors. However, even the majority of tumor cells of early-induced tumors progress over time to a Syp^+^Imp^-^ state. Only a small fraction of tNBs express Imp at this time (Narbonne-Reveau et al., 2016). Rather than influencing the temporal state of the NB in which the tumor is induced, we suggest that cherub regulates temporal fate transitions in tNBs. Loss of cherub enhances the progression towards an “aged” tNB with less growth capacity.

12) The model implies that cherub, Staufen and Syp will co-localize in the crescent, but this has not been shown.

We showed that cherub and Syncrip localize at the same cell pole of mitotic NBIIs by measuring the position relative to the apical cell pole (see Figure 6, Figure 6—figure supplement 1 and Figure 9). These results already indicate that cherub and Syncrip localize at the basal cell pole like it has been extensively described for Staufen in the literature (and also Figure 6—figure supplement 2).

To strengthen this point, we now further included immunostainings showing the colocalization of these 3 gene products. Additionally, we also quantified the colocalization by determining the Pearson’s coefficient for NBIIs and performed intensity covariance analysis. These analyses suggest not only that all three localize to the basal crescent, but also that their intensities correlate. All new data can be found in Figure 9—figure supplement 1.